# Decentralized control of insect walking: A simple neural network explains a wide range of behavioral and neurophysiological results

**Malte Schilling**[1]*, **Holk Cruse**[1,2]

**1** Cluster of Excellence Cognitive Interactive Technology (CITEC), Bielefeld University, Bielefeld, Germany,
**2** Biological Cybernetics, Faculty of Biology, Bielefeld University, Bielefeld, Germany

* mschilli@techfak.uni-bielefeld.de

**Data Availability Statement:** Implementations of the model and simulation engine are publicly available: dynamical simulation environment is realized in C++ and based on the Open Dynamics

## Abstract

Controlling the six legs of an insect walking in an unpredictable environment is a challenging task, as many degrees of freedom have to be coordinated. Solutions proposed to deal with this task are usually based on the highly influential concept that (sensory-modulated) central pattern generators (CPG) are required to control the rhythmic movements of walking legs. Here, we investigate a different view. To this end, we introduce a sensor based controller operating on artificial neurons, being applied to a (simulated) insectoid robot required to exploit the "loop through the world" allowing for simplification of neural computation. We show that such a decentralized solution leads to adaptive behavior when facing uncertain environments which we demonstrate for a broad range of behaviors never dealt with in a single system by earlier approaches. This includes the ability to produce footfall patterns such as velocity dependent "tripod", "tetrapod", "pentapod" as well as various stable intermediate patterns as observed in stick insects and in Drosophila. These patterns are found to be stable against disturbances and when starting from various leg configurations. Our neuronal architecture easily allows for starting or interrupting a walk, all being difficult for CPG controlled solutions. Furthermore, negotiation of curves and walking on a treadmill with various treatments of individual legs is possible as well as backward walking and performing short steps. This approach can as well account for the neurophysiological results usually interpreted to support the idea that CPGs form the basis of walking, although our approach is not relying on explicit CPG-like structures. Application of CPGs may however be required for very fast walking. Our neuronal structure allows to pinpoint specific neurons known from various insect studies. Interestingly, specific common properties observed in both insects and crustaceans suggest a significance of our controller beyond the realm of insects.

## Author summary

Insects are able to walk and climb in complex environments, which requires continuous control of at least 18 joints. Thereby insects outperform even modern robots. But while

Engine library, see https://github.com/malteschilling/hector; the neuroWalknet controller has been implemented in python (version 3), see https://github.com/hcruse/neuro_walknet.

**Funding:** This work was supported by the Cluster of Excellence Cognitive Interaction Technology CITEC (EXC 277) at Bielefeld University, which is funded by the German Research Foundation (DFG). The funders had no role or influence in study design, data collection and analysis, decision to publish, or preparation of the manuscript.

**Competing interests:** The authors have declared that no competing interests exist.

robots are built as sophisticated artificial systems, insect behavior is assumed to rely on the interaction of quite simple control principles. Two predominant assumptions are that insects use–for coordination between legs–discrete gaits, and–as a basis to coordinate the joints of a leg–neuronal rhythm generators. As application of these principles allows description of only a limited amount of behavioral data, both assumptions are challenged here. First, there are no discrete, separate gaits. Instead, there is a continuum of emergent leg patterns as has been known since long for stick insects and recently also confirmed for Drosophila. Second, concerning the control of different joints of a leg, we argue that, apart from very fast walking, neuronal rhythm generators are not required, but may rather be counterproductive as concerns computational efficiency. Instead we propose a decentralized, embodied neuronal structure exploiting sensory feedback and dynamic switching between internal states. This system explains data provided by a large amount of behavioral and neurophysiological studies as well as basic aspects of different species.

## Introduction

Motor control in animals deals with the control of a very high number of muscles (or degrees of freedom) that have to be controlled in a coordinated way in order to achieve complex behaviors. One key principle that allows animals to deal with such a complex control problem is modularization [1–4], i.e. breaking the overall complexity of the control system down on a structural level [5]. Research on locomotion in particular has highlighted decentralization as an organizing principle of such a modular structure that leads to parallel, local processing which is highly efficient, but requires some form of coordination.

Locomotion is a basic behavior of animals, the most important types of which are swimming, flying and walking, behaviors preponderantly characterized by rhythmic movement [6]. Locomotion has been studied in insects for a long time as this offers insights on different levels of analysis that are assumed to translate to other animals as well [7,8]. Insects, on the one end of the spectrum, allow studying neuronal activities on a detailed neurophysiological level [9–12]. On the other hand, insect behavior can be observed in specific and well-defined contexts [13–16]. As a result, decentralization has been established as one key organizing principle, but there is disagreement on the internal organization of the local control modules and in particular if locomotion is driven by an intrinsic oscillation as such or by sensory input signals [10,16–25]. Rhythmic biological movements–as long as these cannot be explained by mechanical, e.g., pendulum-like, properties–are generally considered to be controlled by intrinsic oscillations that are driven by neuronal systems and provide the basis of the required motor output (e.g., [6,20,26,27] a view strongly supported by studies on various types of these animals. But many of the cases studied concern swimming or flying or running. The situation seems to be less clear when it comes to a different type of locomotion, namely walking. What might be crucial conditions that make a difference between (slow) walking on the one hand and swimming or flying on the other? The main difference appears to be given by the different environmental situations animals have to deal with [10,13].

Swimming and flying are operating in a fairly homogenous, fluid media. Walking, in contrast, has to deal with a more difficult substrate, providing unpredictable situations with discontinuities leading to strong mechanical accelerations and which may vary dramatically from step to step. In contrast to swimming and flying, walking is characterized by the necessity to orchestrate the movement of many joints (see below). To cope with this problem, evolution has designed legs characterized by specific morphological properties [28]. Each leg has several

joints, at least three, in insects often four, in crustacea even more, leading to a high number of degrees of freedom to be controlled. As a result, at least 18 degrees of freedom (DoF), three per leg, exist in the case of a hexapod. 12 of these represent redundant degrees of freedom. This high number means that the kinematic system is characterized by "superfluous", or redundant DoFs, which allow for highly adaptive behavior, but at the same time complicate the control problem [29]. The controller of such an underdetermined system is further challenged by the fact that these joints are not completely free to move, but are constrained by mechanical limitations among body and substrate making the control task even more complex [30].

In the remainder of this article, we will concentrate on walking of arthropods, with a focus on hexapods and will begin with a behavioral level of analysis, highlighting the fact that walking comprises a behavioral spectrum with respect to different velocities that pose quite different challenges to the system. Later in the Introduction, we will turn towards related neurophysiological research that raises the question as to how intrinsic oscillatory systems may contribute to control of walking. The goal of this work is to provide a novel control model that for the first time attempts to cover the full range of these data.

## A continuum of locomotor behavior

Insect locomotion on solid surface spans a whole behavioral continuum [7,31]–from slow walking to running including exploratory stepping, gap crossing, climbing and turning. The contexts of the ends of this spectrum differ quite a lot. On the one hand, a typical and well studied example for slow walking can be found in the stick insect [10,32–34]. Stick insects normally climb through bushes and try to hide in twigs by mimicking them. They are moving slowly–less than three steps/s–as required by their behavioral context. When climbing through twigs there is no continuous substrate for walking. Instead, an insect has to search for footholds. During walking, antennae (and front legs to a certain degree) perform searching movements to sample the environment for footholds [35–37]. Furthermore, once a foothold has been found this information is shared with other legs that later-on will target and reuse that same foothold again [14,15]. In this case, the environment is dictating the conditions where to place a foot and in this way also the length of individual steps, which means that the stepping pattern may be quite variable. Temporal stability concerning the stepping pattern is less important in slow walking because longer stance duration allows for higher variability [23]. Questions concerning postural stability are even less critical during slow walking as the insects can grasp the substrate with their tarsi during climbing. To summarize, slow walking behavior is characterized by quite irregular coordination patterns as a consequence of the constant adaptation required by the given context.

At the other end of the spectrum, there is a large body of research on very fast walking, sometimes termed running, in insects [13,25,38,39]. Here, the situation differs considerably: For example, in cockroaches step frequency can be very high, up to 12–15 steps per second in *Blaberus discoidalis* [11,13,40]; for *Periplaneta americana* even 25 step/s have been reported [16,38]. For such very fast movements the delay of sensory pathways does play a role. Sensory information flow appears too slow to drive behavior selection [11]. Instead, the control principle has to rely on an (in insects probably implicit) estimate of the continuation of the movement. The behavior is driven intrinsically which is supported by properties of the body [41–43]: postural stability becomes more a question of dynamic stability and elasticities in the legs can support and stabilize the behavior as they are, for example, mechanically counteracting small disturbances in the substrate [44]. To summarize, running is characterized as a rhythmic activity that leads to quite stereotypical coordination patterns of the legs being supported and stabilized by properties of the body.

To differentiate running and walking we want to consider which stepping frequency might represent a limit above which sensory influence is not effective anymore? Sponberg and Full [11] argue that for *Blaberus discoidalis* the fastest reflex response requires between 16 and 20 ms and assume that this delay is already too much to control switching between stance and swing states for a frequency of 10 steps/s (i.e., a period of 100 ms) or more. Results of Delcomyn [45] with *Periplaneta americana* show that sensory feedback plays a role below 5 Hz but not anymore above this line. This means velocities up to 5 steps/s (period 200 ms) may be controlled by sensory feedback. Data of Zill and Moran [46] (*Periplaneta americana*) suggest a limit of 7 steps/s (for *Drosophila* this limit might be higher due to its small size and therefore shorter delays). These data should be considered as estimates only, as conduction delays from sensors on proximal leg segments can be very short and could function at higher step frequencies [47]. Another possibility as to how "running" may be controlled, but not considered here, is that sensory input in one phase could influence the output of the next phase (pers. comm., Sasha Zill). It is an open question whether "running" should be considered as a separate gait applying a quite specific pattern [13,40], and if so, how a distinction between walking and running may be defined. In the following we will use the term "running" if, for a cockroach, the step frequency is beyond about 7 steps/s. In this article, our focus is on a controller structure, neuroWalknet, that covers only patterns in a range that allows sensory inputs being effective (see however Discussion and Supporting information).

Fast walks, also below 5 steps/s are generally characterized by using a tripod pattern (i.e., at least three legs support the body at any moment of time). But tripod patterns can be observed in "slow" stick insects, too. In fact, there is a continuum [23,34] ranging from very slow walking ("pentapod" pattern sometimes called "wave gait", at least five legs are on the ground at any time) over "tetrapod" pattern, sometimes called "metachronal gait" (at least four legs on the ground) to tripod pattern (at least three legs are on the ground). Using these terms for different walking patterns seems to imply that there are specific, discrete "gaits" as known from horses, for example. When we, nonetheless, use these terms, as we do for "running", this is done for practical reasons and should not be understood in the sense of discrete gaits as observed in mammals. In cockroaches, *Periplaneta americana*, below about 3–4 steps/s a tetrapod gait is observed [13,48]. In the stick insect *Carausius morosus* this transition occurs in a similar range (about 1–2 steps/s [34]). Stressing these differences might lead to the–unintended–assumption that we are merely looking at different phenomena–different behaviors with different underlying control schemes. However, this is not necessarily the case. Rather, these results suggest that for forward walking there is a continuum of intermediate stable stepping patterns observed in possibly all insects, at least in cockroaches and in stick insects, both representing the group of hemimetabolic insects, as well as in *Drosophila* [49], representing a holometabolic insect. Recently, this view has been strongly supported by a study of DeAngelis et al. [50].

## Free gait controllers: Emergence of coordination patterns

We now want to turn towards a perspective that is focusing on underlying mechanisms. How may a controller be structured that is able to deal with the characteristics mentioned above? One assumption might be that specific behaviors observed may require specific neuronal modules (or motor schemas, see [51]) which can be separately switched on or off (instructive examples are given by Steingrube et al. [52] who distinguish eleven types of behaviors, seven of which are different walking types, or Daun-Gruhn and Tóth [53] who use separate networks for tripod and tetrapod gaits).

However, as the behavioral data proposes that we deal with a continuous behavioral spectrum, this concept does not support a view of distinct motor schemas on the neuronal level ([34] for *Carausius*; [49] for *Drosophila*; see also [54] for discussion). Discrete "gaits" only appear in the eye of the observer and it might therefore be misleading to assume that a controller has been evolved to produce separate gaits (as mentioned, the situation may be different for running). Instead, it appears plausible that the controller has been designed to deal with general, unpredictable environments including unexpected disturbances, and that natural patterns strongly depend on specific environmental conditions (e.g., walking speed, or geometrical properties of the environment, for example when climbing over large gaps or along a vertical branch), where the "loop through the world" and feedback from the environment is assumed to play an important role [55]. Similarly, straight forward walking or negotiating curves might not result from two separate modules ("straight forward" or "turning"), but may result from a system able to control turning by using various curvature radii [56–60] which includes straight forward walking. However, backward walking may require a separate state. For further examples showing that not every behavioral phenomenon that we recognize requires a specific and explicit neuronal module, see [30,61–63] and our Discussion.

Therefore, starting the design of a controller for the observed behavior by introduction of, at the outset, distinct structural elements might appear sensible in some cases, but should be avoided as long as no strong arguments support such a separation.

## Decentralization as a central control principle

To cope with the ability to walk, two problems have to be solved (e.g., [15]). One (A) concerns the question of how to couple the movement of the different legs and the other (B) is how to control the movement of an individual leg. For both problems modularization appears to be a sensible solution [2,64].

(A) The long tradition of behavioral research in insect walking has contributed largely to the understanding of how to couple the movement of different legs. Interleg coordination rules have been derived in diverse experimental settings that structure the overall behavior and coordinate the individual leg controllers. The whole spectrum of different gaits in a wide variety of contexts appears to emerge from such control structures. One example is given by our Walknet approach [23,24], which is based on individual leg controllers and coordination influences that mainly regulate the length of the step size. This system has been successfully applied in diverse contexts in dynamic simulation and on many robots. Whereas in Walknet only three rules were applied, in neuroWalknet we will also apply rule 5 [15]. In short (for details see Methods), rule 1 acts between ipsilateral legs and inhibits start of swing in the anterior neighboring leg during swing of the sender leg. Rule 2 stimulates swing of the anterior or the contralateral neighboring leg if the sender has just finished swing. Rule 3 elicits swing of the anterior neighbor or the contralateral leg, but this signal depends on the leg position of the sender leg (for dependence on velocity see Methods). Rule 5 becomes active if a specific agonist branch is activated beyond a given threshold. Via rule 5 the corresponding agonists in the directly neighboring legs are stimulated. This stimulation is excitatory for all legs except for the two hind legs, where the stimulation is inhibitory (for more details concerning interleg coordination see discussion in Supporting Information S3 PDF).

(B) Another intensively discussed question on this level of organization concerns how to control the behavior of an individual leg, i.e. how to select one of (at least) two behaviors: producing a stance movement in order to support and propel the body or a swing movement, during which the leg is lifted off the ground and moves into the opposite direction to start a new stance. As addressed above, the behavioral evidence may provide different answers depending

on the context and the velocity of walking. On the one hand, slow walking requires integration of detailed feedback from the environment. On the other hand, very fast walking, or running, cannot keep up with integration of sensory input due to sensory delays. In this context, the controller has to produce a rhythm on its own. To cope with these different requirements, it is generally assumed that production of intrinsic rhythm represents the basic function and that such systems are, during slow walking, influenced by sensory feedback to allow adaptation to properties of the environment (for reviews see [17,65]). Here we show that a slightly changed concept can explain more data by assuming that in the intact animal intrinsic rhythms are only applied during running. Our model shows that a form of mutual inhibition as a key organizing control principle is sufficient to explain, on the one hand, sensory switching of behaviors on the single leg level, and on the other hand, that more stable gaits and (seemingly) fixed coordination patterns emerge when the same structure is driven with a different velocity. Nonetheless, this controller is able to deal with the full spectrum from pentapod to tripod walking, without relying on intrinsic rhythm generators. Beyond a given step frequency different connections may start to dominate (and in the end completely override) sensory feedback.

Both sides of the discussion that concerns either intrinsic rhythms or sensory-driven control often focus on quite specific experimental settings. Therefore, in order to show that the proposed control structure addresses the whole spectrum, it has to be tested in all of these contexts. Thus, in the next section we briefly summarize the view of intrinsic rhythms which largely puts a focus onto a different level, as are detailed neurophysiological experiments.

## Central pattern generators

As mentioned above, intrinsic rhythms are often assumed to be caused by central pattern generators (CPG) ([6,20], for a clear definition of the CPG concept see [18,22]). The recognition of CPGs as the basic control system is based on experimental findings made by studies of deafferented animals [6,8,66]. Deafferentation is realized operationally by interrupting sensory input and motor output and later, in addition, by stimulating the neural system by, for example, application of pilocarpine (e.g., [67]). As detailed below, such experiments could demonstrate that motor neurons driving antagonistic muscles of a given joint show–usually low frequency (in the order of 0.2 cycles/s)–oscillations with some similarity to those shown in intact moving animals. As these studies cover a broad range of species, both vertebrates and invertebrates, these results have led to the broadly acknowledged assumption that such CPGs are responsible for controlling the basic rhythm observed in locomotion [6,66], see however [18,22]. In addition, sensory feedback is assumed to be required for modulation, for example for resetting these CPGs allowing for adaptation to varying environmental conditions [10,16].

In the cases of swimming or flying, the temporal coordination between different CPGs observed in the deafferented insects shows in-phase coupling between contralateral neighbors, which correspond relatively well to coupling found in the intact, behaving animal (see below). This is in contrast to the case of walking behavior that is characterized mostly by anti-phase coupling of contralateral neighbors [34,38,49]. Here, the coordination of deafferented leg controllers is often quite different, eventually even opposite to that of normal walking (e.g., [54,67–69], we will deal with an interesting different case [70,71] in the Discussion and Supporting information).

This situation is highlighted by a detailed recent study on locusts [69], focusing on interleg coordination in deafferented locusts (recordings from motor neurons activating the depressor muscle). The authors argued that the temporal patterns observed in the different thoracic ganglia may provide the basis for that found in walking animals. However, coordination in walking locusts is quite different to that found in the deafferented, in situ preparations. In general,

in fully or partly deafferented insects ipislateral neighboring legs fire in-phase, whereas in walking these legs typically show about anti-phase coupling. Supporting results for *Carausius* are presented by Mantziaris et al. [54], and Borgmann et al. [68,72], who recorded from the motor neurons activating the retractor muscles, and by Büschges et al. [67], who recorded from motor neurons of all three joints. Interestingly, corresponding results have also been found for crustacea [73,74]. Furthermore, Knebel et al. [69] found in-phase coupling also between contralateral legs if all three thoracic ganglia were treated with pilocarpine, which again contrasts to normal walking behavior, but can be observed in swimming ([75], crayfish swimmerets) and flying ([8], locust). These results raise questions concerning the contribution of these oscillatory systems to control of walking.

### Simulations as a functional tool to analyze principles in a behavioral context

Different to Walknet [23,24] our current controller "neuroWalknet" is realized by an artificial neuronal network with neurons endowed with dynamic properties. Furthermore, the motor system is organized using an antagonistic architecture [53,76–79] as such a more fine grained neuronal anatomy may allow for more detailed interpretations of the control structure proposed. Our goal is to search for a simulation that allows to better understand specific biological systems, in particular, the stick insect *Carausius morosus* (see Fig 1).

For our approach, the simulation requires a body, legs, actuators for each joint, sensors, and a physical environment. The latter is important, because the existence of a body with legs coupled via the environment may allow simplifying neuronal computations [55]. Concerning simulation of the body, software simulation and hardware simulation may be used. Here, we start with a dynamic software simulation of the robot Hector (Fig 1B, [80,81]). With such an architecture, the following general questions will be addressed.

(i) How may (velocity dependent) interleg coordination [15] be realized on the neuronal level in a way that temporal patterns will emerge that correspond to the patterns observed in behavior and that can adapt to external disturbances?

(ii) How could the three joints of a single leg be coordinated? The joints of a leg are often not moved in-phase, but, depending on the current environmental situation, may show varying phase values. Neural motor output and torque produced may not be directly coupled [82]. Negotiation of curves adds to the complexity of the task. Various solutions for intraleg

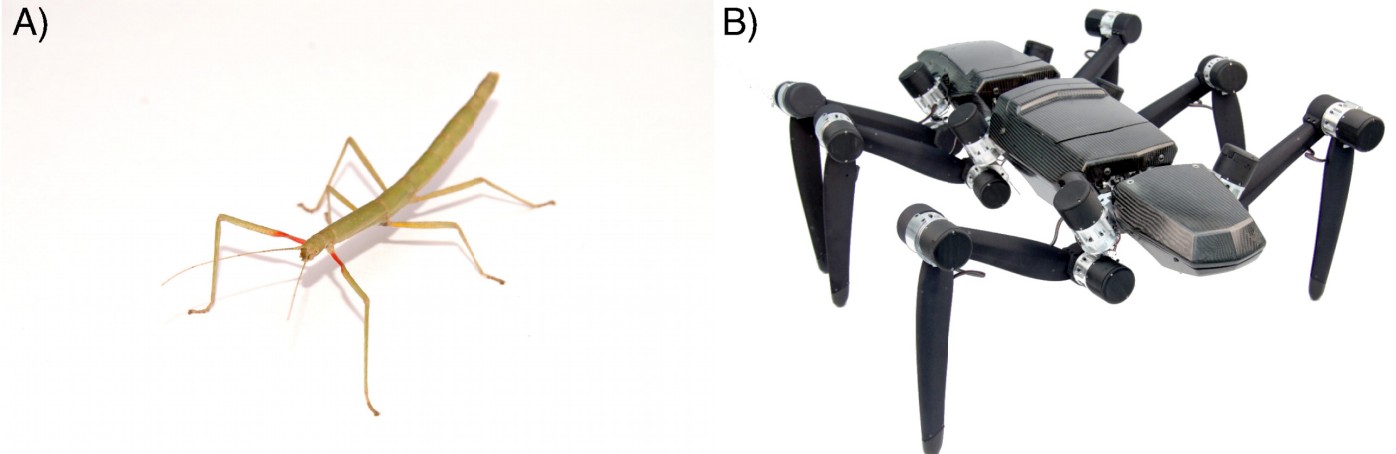

**Fig 1.** (a) Indian stick insect *Carausius morosus*. (b) Robot Hector which is used in simulation for testing the control principles derived.

coordination have been proposed [52,53,77,83], but have not been tested in the full range of experimental data available.

(iii) Is the principle of mutual inhibition as applied in the proposed architecture sufficient to explain intrinsic oscillations that emerge in specific experimental situations (e.g., [69,72,73])?

Overall, the goal of the approach is not to reproduce specific, singular observations looked at in isolation, e.g. a specific gait type. In contrast, our goal is to establish a controller, a holistic model based on a minimum of a priori assumptions, that is able to describe a large amount of known behaviors that emerge from its decentralized structures [23]. These behaviors concern the full continuum of stepping patterns as observed in insects from pentapod via tetrapod to tripod patterns including intermediate stable patterns [34,49], backward walking [84,85] and negotiation of curves [57,59,60] as well as stability against disturbances of leg movements studied in many experiments (for a review see [14]). Furthermore, a number of neurophysiological studies should be described that demonstrate the appearance of intrinsic oscillation also in slow walking arthropods, both insects and crustaceans (e.g., [69,72,73]) which do, however, not contribute to normal, slow walking. Such a simulation may provide a holistic description, suited for testing hypotheses as well as to stimulate new experimental approaches. Note that this is not meant to neglect the general existence of slow CPGs to control rhythmic behavior. We only ask whether such CPGs are necessary components for slow walking.

## Results

The proposed neuronal architecture (for details see Methods) has been tested in a variety of different experimental situations, ranging from diverse behaviors to analyses of neural activity. It follows the basic characteristic of Walknet [23,24], but is now constructed using a detailed neuronal architecture (Fig 2 and Methods). Central to this control approach is the notion of decentralization realized as individual controllers for each leg. Front legs, middle legs and hind legs are controlled by a ganglion each, as found in the stick insect: the prothoracic ganglion, the mesothoracic ganglion and the metathoracic ganglion, respectively. Each ganglion consists of two hemiganglia, right and left, which control the corresponding legs. Interleg coordination is organized by local rules acting between neighboring legs. On the one hand, a set of these rules influences step length (coordination rule 1, 2 and 3, represented in Fig 2 by brown units, see also Material and Methods). On the other hand, as an additional rule that has been introduced into the Walknet control structure [86] rule 5 (Fig 2, shown in ocher) is incorporating load-dependent signals into the controller. To allow for running, a further rule is explained in Supporting information.

The movement of a walking leg is characterized by two states, stance and swing. Following the antagonistic structure of the biological motor system the output of the individual joint muscles is controlled by an antagonistic structure, too. Each muscle is driven by a motor neuron (Fig 2, leftmost red units, MN). To minimize co-contraction of antagonistic muscles, premotor units (Fig 2, red, right neighbor of each motor unit, PMN) are connected via inhibitory units forming a four-unit recurrent network. The input to the controller (apart from local sensory feedback, Methods) is given by one unit (Fig 2, upper part, below the brown units representing coordination influences, leftmost white unit, input "vel", in the following termed velocity neuron") that represents the desired global walking velocity (given in mV between 0 and 50 mV). The output of the controller, i.e. the motor neurons (MN), is a velocity signal that is used to drive the joint (for further explanation see Methods). This controller architecture is assumed to encapsulate quite general control principles across different animal species [7].

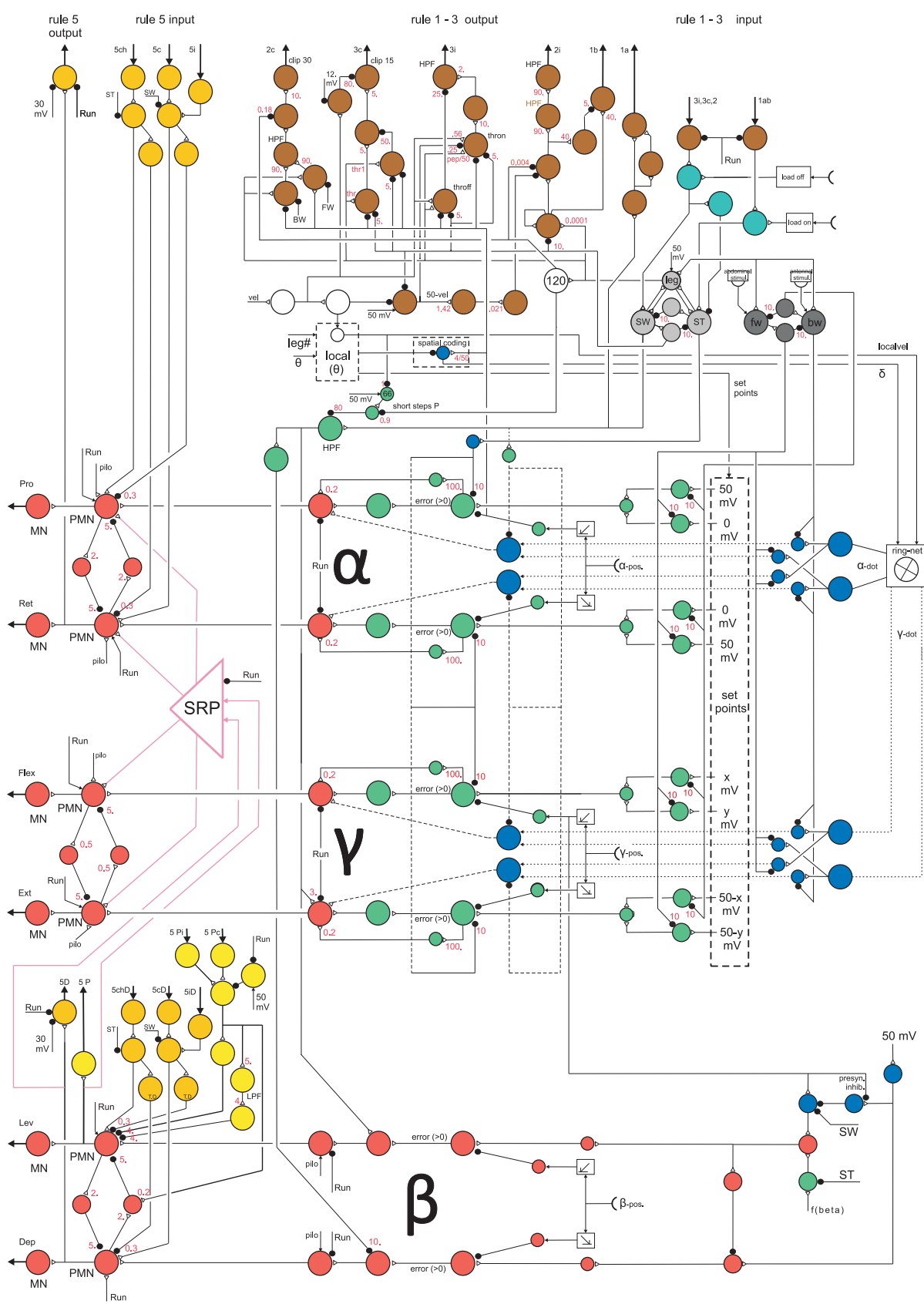

**Fig 2. Controller for one leg.** Artificial neurons are depicted as colored circles. Sensory input structure for swing: green, sensory input structure for stance: blue. Common output: red. Coordination channels: brown for rules 1–3, ocher for rule 5, light yellow for rule P (Pearson rule, see Supporting information) with connections in pink (Supporting information). Motivation units: light grey (swing–stance), dark grey (forward–backward), white (central units). SRP net and its' connections (pink) are detailed in Supporting information (S2 PDF). Joint alpha (α): Pro–protractor, Ret–retractor; joint beta (β): Lev–levator, Dep–depressor; joint gamma (γ): Flex–flexor, Ext–extensor. Black dots: inhibitory synapses, open black triangles: excitatory synapses. For detailed explanations see text and Supporting information.

In order to show the reach of these principles, the controller is applied and tested on a scaled up robotic model of an insect even though the dynamics in this scaled version are changing in a non-favorable way. As an insect, the model has three pairs of legs, front legs, middle legs and hind legs (FL, FR, ML, MR, HL, HR, with L for left and R for right leg). Each prototypical leg is equipped with three joints, termed alpha joint, beta joint and gamma joint. The alpha joint can be abstracted to be driven by two antagonistic muscles, protractor and retractor, essentially moving the leg from rear to front and back, the beta joint, with two antagonistic muscles levator and depressor, moving the leg up and down, and the gamma joint with two antagonistic muscles termed flexor and extensor, which essentially decrease or increase the length of the leg.

To illustrate the simulation results, apart from videos (Supporting information S1–S8 Videos show forward walking at different velocities, S9 Video running, S10 Video curve walking, and S11–S14 Videos backward walking), so called foot fall patterns will be used: The temporal orchestration of the six legs are depicted by black bars showing the swing state (Figs 3, 4, 6 and 7, Supporting information S2 Fig, S3 Fig). In other cases (legs being deafferented) black bars illustrate the activation of specific motor neurons when above a given threshold (Figs 8–10, Supporting information S4 Fig and S2 PDF).

In the following, first, behavioral results will show that neuroWalknet can produce diverse walking behaviors: different gait patterns emerge from different velocities, the system can negotiate curves and can deal with backward walking. Second, neural activities of the control structure will be analyzed in detail in order to compare these results with the findings of intrinsic oscillations as found in various neurophysiological experimental settings, for example dealing with deafferented animals.

## Forward walking–emerging gaits

Which patterns are produced when neuroWalknet is walking using all six legs but at different velocities? Fig 3 shows footfall patterns for different velocities, ranging from fast to very slow walking (Supporting information S1–S8 Videos show videos). Velocities are fed into the system as an external input given to the velocity neuron (Fig 2, leftmost white unit). Here, inputs of 50 mV, 40 mV, 30 mV, 25 mV, 20 mV, 15 mV 10 mV and 8 mV were used (we also tested 45 mV, 37.5 mV, 35 mV and 32.5 mV, see Supporting information S2 Fig, for corresponding videos see Supporting information). All walks are shown for 80 seconds. In the simulations shown, the robot always started from the same posture. As this was close to a tripod pattern, for higher velocities this posture already fitted the temporal pattern quite well, and accordingly a stable pattern was reached very soon for velocity signals induced by 50 mV and 40 mV. This starting configuration was however not optimally suited for most of the slower walks. Nonetheless, stable patterns emerged fast for velocity inputs down to 20 mV and after only a couple of steps for 15 mV. In the other cases (velocity neuron set to 10 mV or 8 mV) more steps were required to reach a stable state. Fig 3 (velocity neuron set to 10 mV) shows a case in which a stable pattern was reached quite fast for a slow velocity due to introduction of an additional disturbance (a deliberate prolongation of the first swing of the right middle leg). Fig 3, velocity set to 8 mV, shows a stable state reached only after more than 240 s. The transition patterns

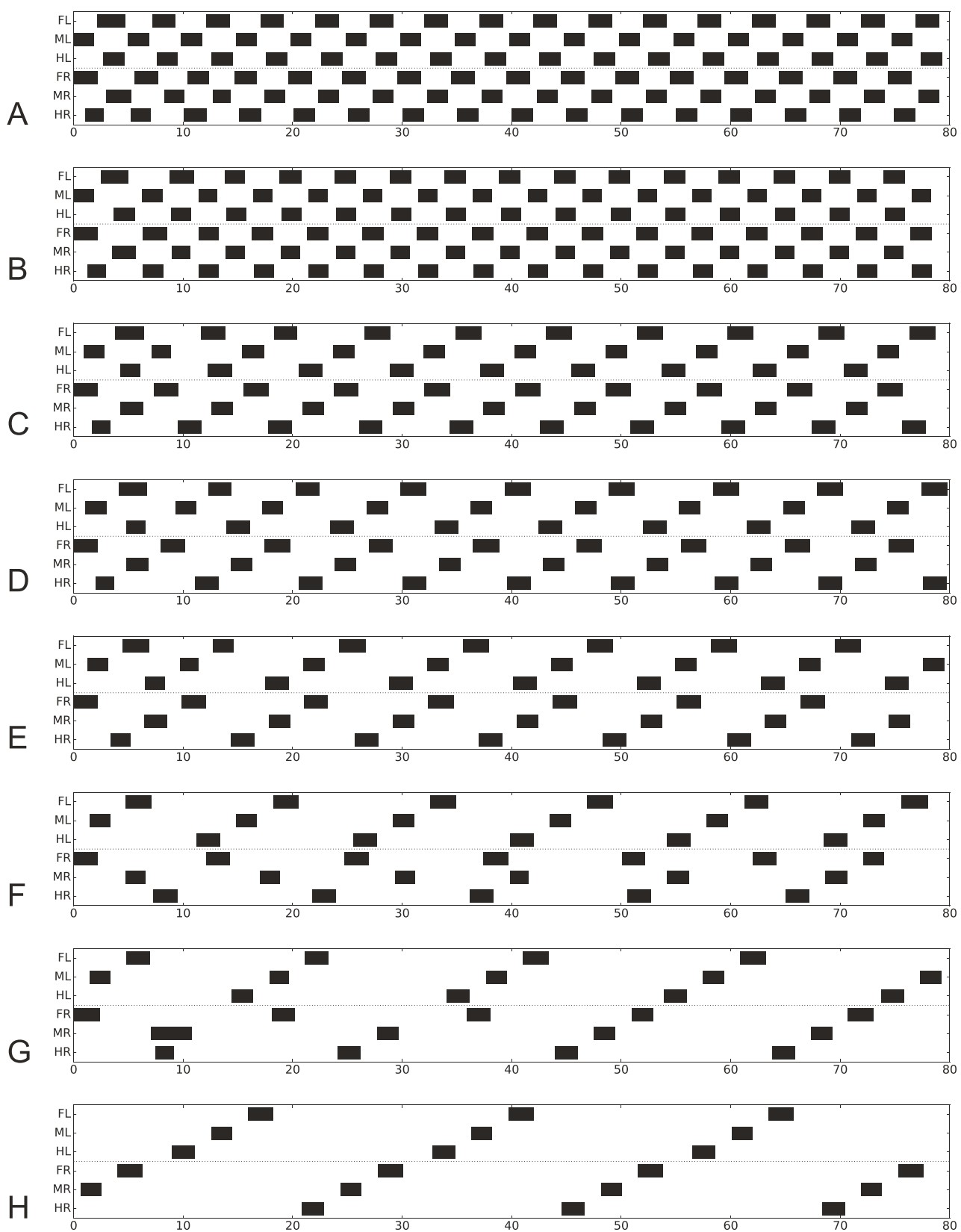

**Fig 3.** A-H. Footfall patterns for velocities driven by 50 mV, 40 mV, 30 mV, 25 mV, 20 mV, 15 mV 10 mV and 8 mV (G: the latter after disturbance of MR at about 8–11 s, see prolonged swing) and 8 mV (H: a stable pattern reached after more than 240 s). Apart from the last two runs all started with the same leg configuration. The corresponding transient phases for the last two runs are given in Supporting information (S3F and S3G Fig). Black bars show swing mode, Legs: FR front right, MR middle right, HR hind right, FL front left, ML middle left, HL hind left. For videos see Supporting information S1–S8 Videos. Abscissa: Time (s).

found when using the generally applied starting configuration without additional disturbances are depicted in Supporting information for 240 s each (S3 Fig).

Using the traditional characterization, patterns driven by velocity input of 50 mV or 40 mV correspond to tripod gait, patterns produced between 30 mV and 15 mV are obvious tetrapod gaits, whereas patterns produced around 37.5–35 mV or between 15 mV and 10 mV (see velocity neuron set to 12.5 mV, Supporting information S2 Fig) may be termed intermediate gaits, or perhaps tetrapod, depending on the exact definition. Patterns produced by velocity neuron set to 10 mV and below correspond to pentapod gaits. These results show that stable patterns emerged from the control structure and the interaction with the environment showing adaptive behaviors that can deal with disturbances given by uncomfortable starting configurations or deliberate manipulation (see Discussion). In the simulation, this continuum results from the fact that the ipsilateral patterns essentially depend on stance duration which in turn strictly depends on velocity.

Whereas typical tripod gait patterns and wave gait patterns are symmetrical with respect to right and left legs, this is not the case for the group labeled tetrapod. Here, for a given velocity,

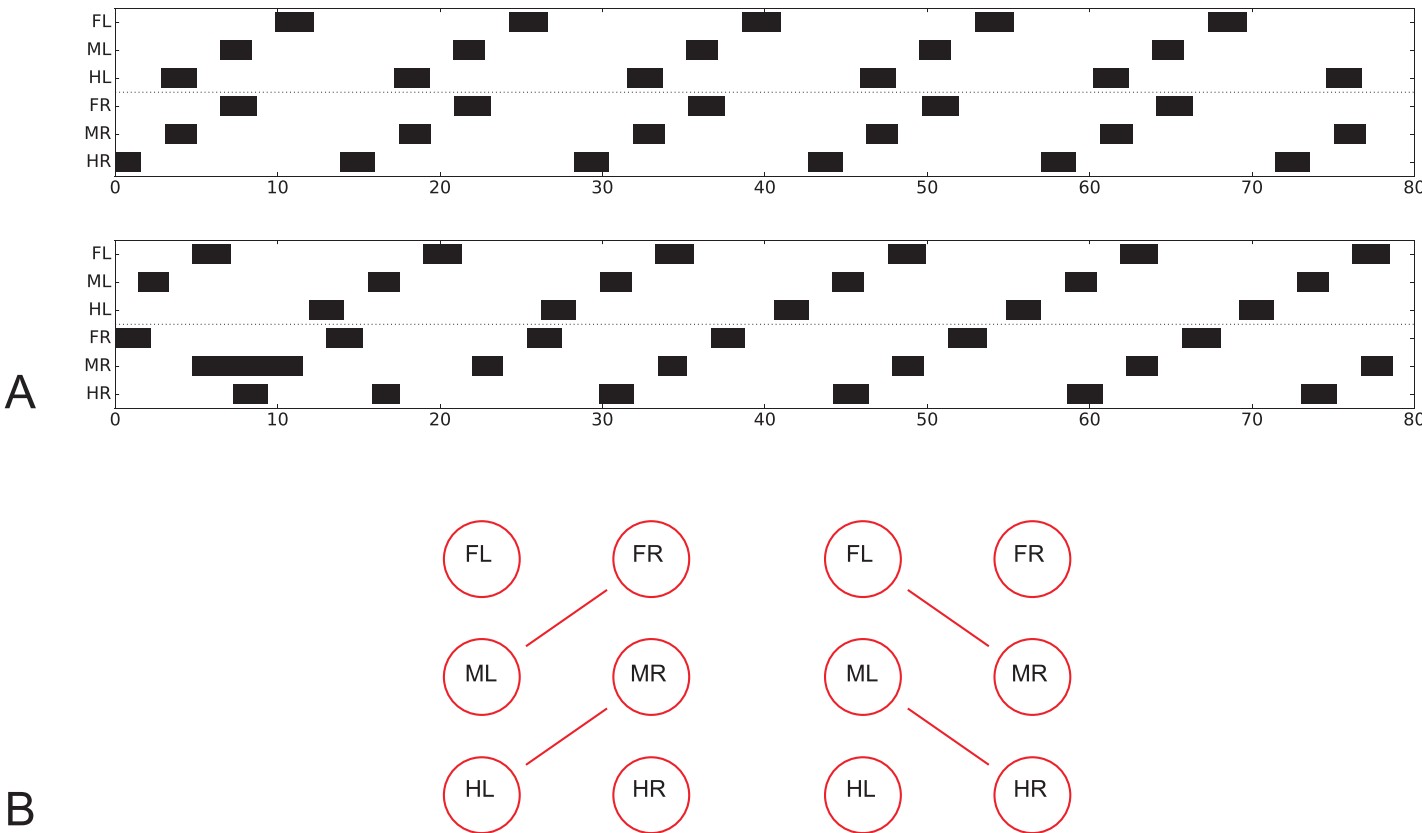

**Fig 4. Footfall patterns with the velocity neuron set to 15 mV.** Upper panel (A) as in Fig 3F, after a stable pattern has been reached; lower panel which, after a disturbance of the right middle leg during swing, represents a mirror image version. B) illustrates which leg pairs are coupled together in either case.

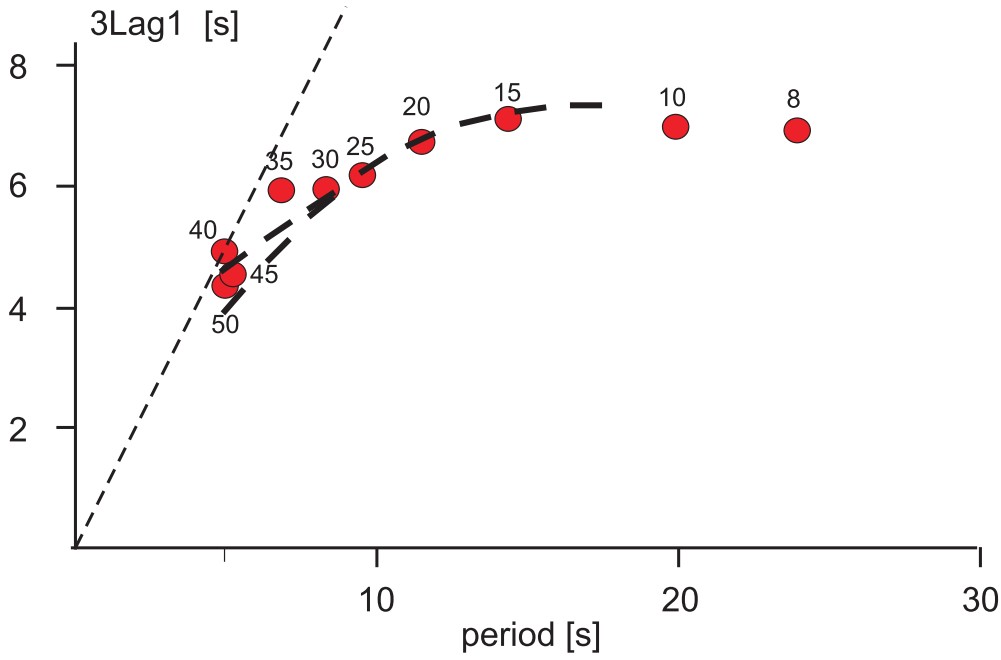

**Fig 5. Duration of lag 3L1 (time lag between beginning of swing in hind leg and beginning of swing in the following front leg), vs. period (duration of swing plus stance) as proposed by Graham [34].** Numbers attached to the red dots mark the velocity input (mV) given. Abscissa: time (s). The thin dashed line indicates slope 1. Bold dashed lines show the average data from Graham [34], his Fig 7; time *0.1). For short periods Graham observed a slight asymmetry between left and right legs, not observed in our simulations.

two mirror image versions are theoretically possible. Indeed, both versions can be produced when the walker is disturbed in some way (or, correspondingly, another starting configuration had been used). This is shown for velocity neuron set to 15 (Fig 4), where Fig 4A, upper panel, shows the pattern directly following the undisturbed version as presented in Fig 3, whereas Fig 4A, lower panel, shows a walk with the same starting configuration, but a disturbance applied after about 15 s by artificially prolonging the swing duration of the right middle leg. Fig 4B illustrates which leg pairs swing together in either case. Note that these patterns show a contralateral phase different from 0.5, which agrees with experimental findings observed during tetrapod gait (*Carausius*: [34]; *Drosophila*: [49]). For higher and for lower velocities, contralateral phase approaches values of 0.5. Note that averaging mirror image patterns with contralateral phase values of 0.33 and 0.66, for example, would lead to a mean phase value of 0.5, a result that may lead to the misinterpretation that contralateral legs are always coupled by a 0.5 phase, independent of velocity.

To compare the footfall pattern received in the simulation with data obtained from stick insects ([34], his Fig 7), Fig 5 shows original data of Graham (his "3L1", i.e. the time lag between beginning of swing in the (e.g., left) hind leg and the beginning of swing in the ipsilateral front leg vs. leg period). The corresponding values found in the simulation are depicted by red dots. Qualitatively, there is good agreement. Note that time scales differ by a factor of around 10 (this has been adapted as a consequence of the scaling of the robot structure).

Starting or stopping a walk is simply reached by setting the velocity unit from zero to the desired value, or back to zero (not shown). When the walker is stopped, legs that are in swing state finish swing movements, so that at the end all legs adopt ground contact as observed in insects. Controlling start and stop of a hexapod using a CPG-based controller [87] requires a

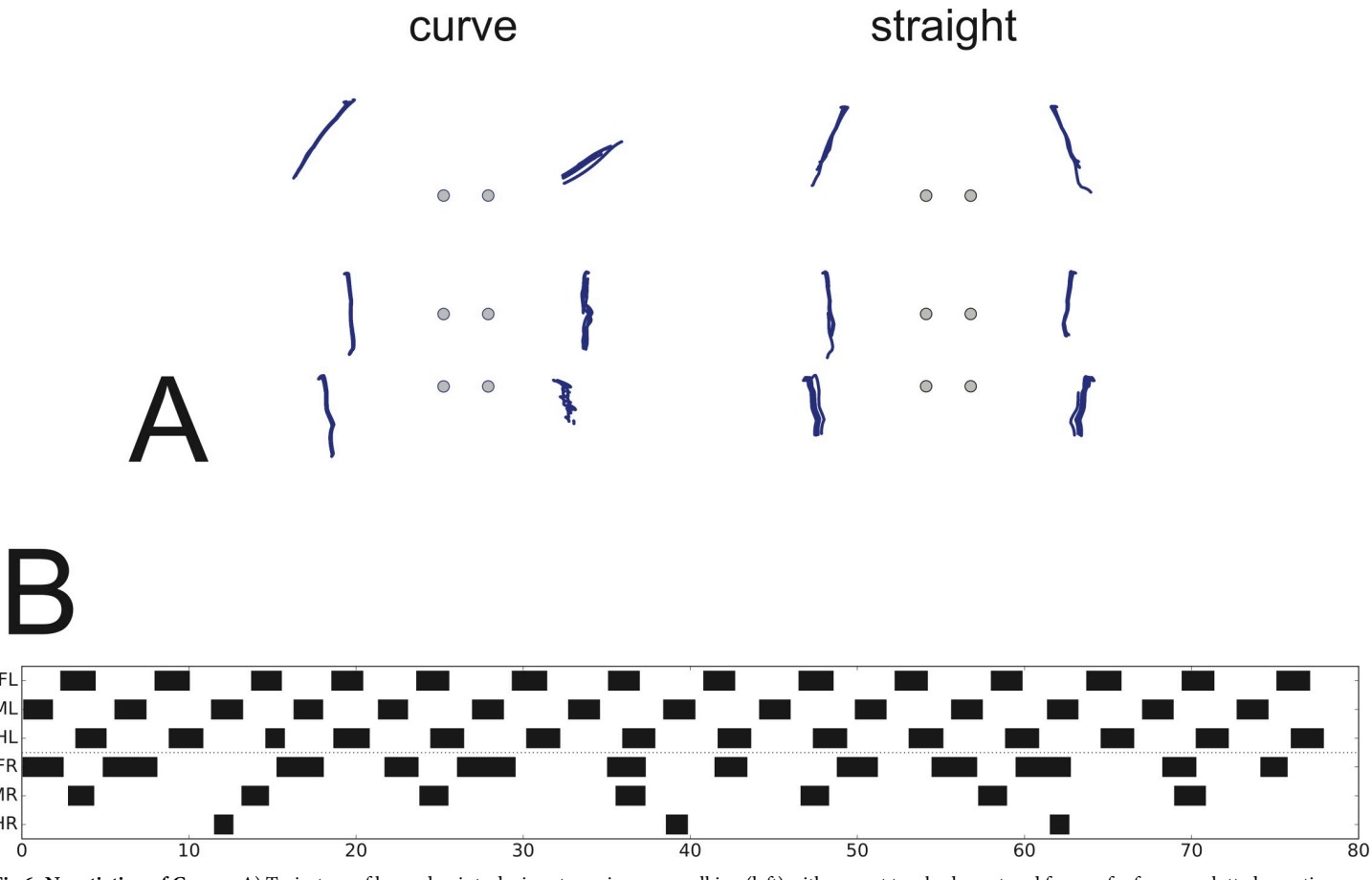

**Fig 6. Negotiation of Curves.** A) Trajectory of leg end points during stance in curve walking (left) with respect to a body centered frame of reference, plotted over time window 50 s – 100 s. On the right, trajectories of leg end points during straight walking is shown for comparison. Dots mark position of leg basis. Front legs up. B) Footfall pattern. Global velocity neuron set to 45 mV, theta 75 deg. Ordinate: legs as in Fig 3, abscissa: time (s). For details see Methods.

considerable effort to tame the dynamic properties of the network. To this end, different CPGs required specific input values for the different leg types while the angular velocities of the beta angles had to fulfill specific conditions. This is an interesting result as it highlights one problem CPG based controllers have to deal with when realistic behavioral tasks have to be solved (related problems are found by [77], too).

## Negotiating curves

An important aspect of forward walking is the ability to negotiate curves [56–60,88]. Dürr and Ebeling [57] show an experiment where simultaneous recordings of footfall patterns, direction of turn as well as AEP and PEP of all six legs have been collected. We focus on this example as a proof of concept as it is the only case where all these data are given for a turning insect. The insect walked tethered on a holder above a sphere and curve walking was elicited visually by a horizontally moving pattern of vertical stripes (angular velocity 38.2 deg s$^{-1}$). While outer legs showed a similar pattern to what is considered a tripod pattern, the inner legs strongly deviated from that pattern. The inner front leg moved in anti-phase to the outer front leg, as is in forward walking. However, the inner middle leg showed an about two times longer step period compared to the inner front leg or the outer legs. The period of the inner hind leg was more irregular and even longer, i.e. showed quite slow velocity during stance. Furthermore, the leg

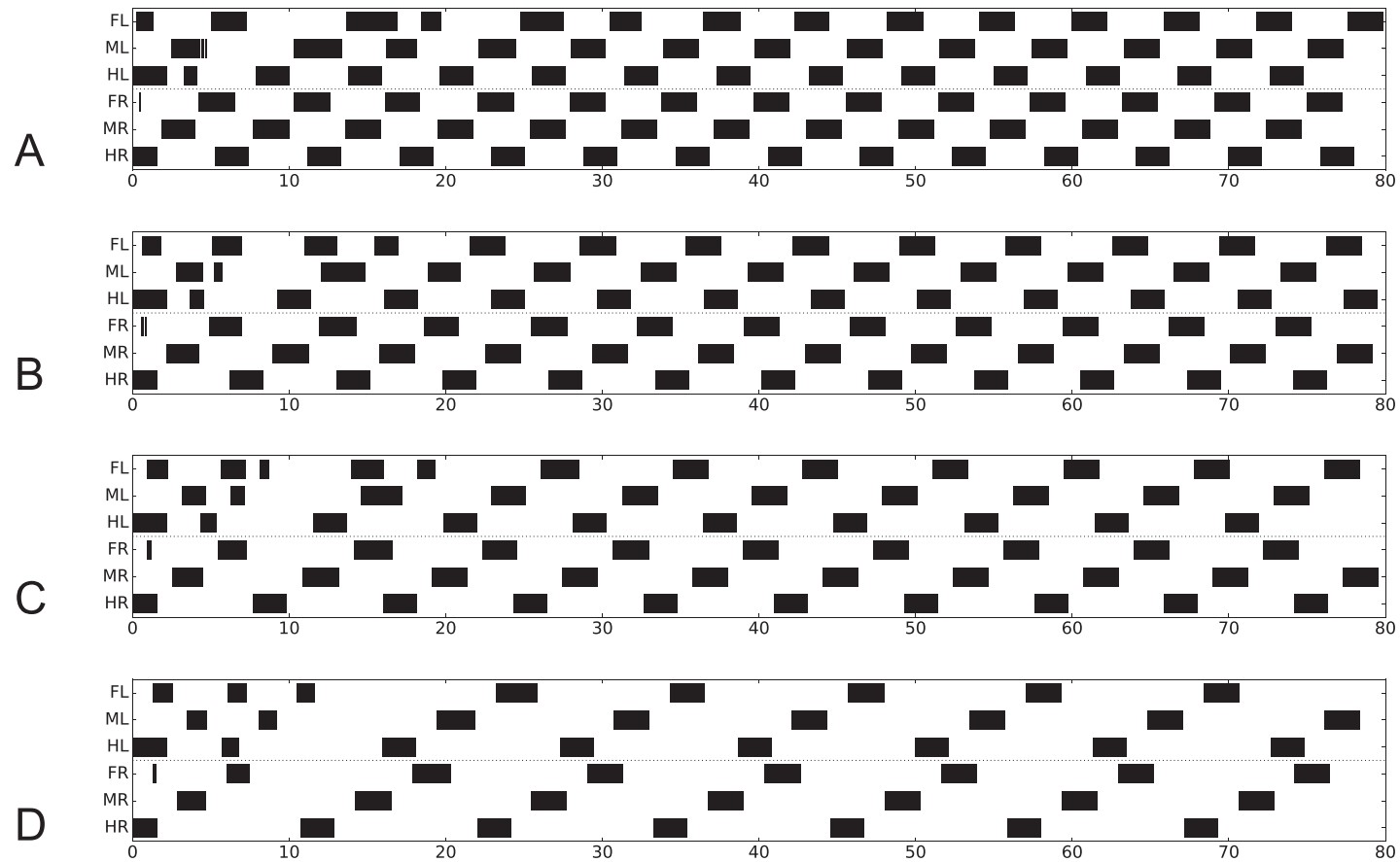

**Fig 7.** A-D. Footfall patterns, backward walking for velocity neuron set to 50 mV, 40 mV, 30 mV or 20 mV (Supporting information S11–S14 Videos). Abscissa: time (s). For further details see Fig 3.

showed very short swing durations comparable to short steps [32,89]. Front legs showed a shift of the AEP in the direction of the desired turning direction, characterized by angle theta [57,59,60,88]. The other legs showed only smaller deviations from normal forward walking. Contribution of front legs was later assumed as sufficient for explaining turning behavior [60].

To test this hypothesis with neuroWalknet, we kept coordination rules as in normal forward walking, but implemented an expansion symbolized by box "local (theta)" in Fig 2, the function of which is to control (i) different set points for swing, (ii) different local walking directions, as well as (iii) different velocity signals for different legs instead of using one global velocity for all legs (for details see Methods).

Results are shown in Fig 6A (Supporting information S10 Video) in a top down view (duration 100 s). A 90 degree turn required about 30 s, the radius of the circle amounts for about 0.8 body length. The footfall patterns (Fig 6B) show qualitative similarity to those found by Dürr and Ebeling ([57] their Fig 2B). The stance trajectories (leg end point trajectories during stance in top view, Fig 6A, left) are similar, too, but there are also clear differences. The amplitude of the inner front leg is smaller compared to those of the animals ([57], their Fig 2Aii and Fig 6B). This is probably mainly due to smaller lengths of the robot legs. Further, in Fig 6A, left, the trajectories of the inner middle leg are parallel to the axis along the body, but are more inclined in the insect (for comparison, Fig 6A right, shows trajectories of straight walking, similar to results of [57]). Such deviations are even stronger when very tight turns are taken [58,90]. This effect is not observed in the simulation because we did not introduce changes of the AEP set

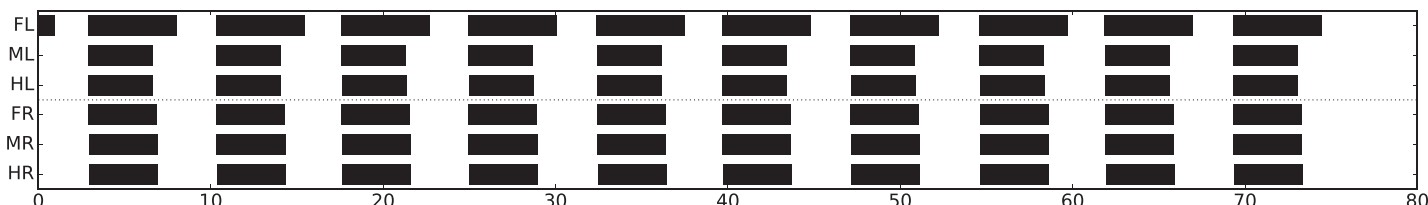

**Fig 8.** A-D. Simulation of experiments of deafferented locusts [69]. Black bars show activation of depressor muscle output (> 0 mV). A) all thoracic ganglia treated with pilocarpine, B) only prothoracic ganglion treated, C) only mesothoracic ganglion treated, D) only metathoracic ganglion treated. Last five periods show stable state. Abscissa: time (s).

points of the middle leg as has been done for the front legs. However, this difference may also be due to the fact that the robot legs may slip during turning which is not the case for insects.

Tests with application of identical local velocities for all legs and changing only the AEP of front legs did not provide curve walking as observed in [57].

Furthermore, in addition to the specific cases mentioned above, systematic disturbances have been performed by prolonging swing states during forward walking. In all cases studied, the controller reached a stable walking pattern (see Discussion, and see also the first steps of backward walking Fig 7).

**Fig 9. Simulation of experiments with stick insects, one leg (FL) walking on a treadmill (velocity neuron set to 30 mV), while the other legs are deafferented and treated with pilocarpine (in Borgmann et al. [72], only ipsilateral legs were recorded).** Black bars show retractor muscle output (> 0 mV). For further velocities see Supporting information S4 Fig Abscissa: time (s).

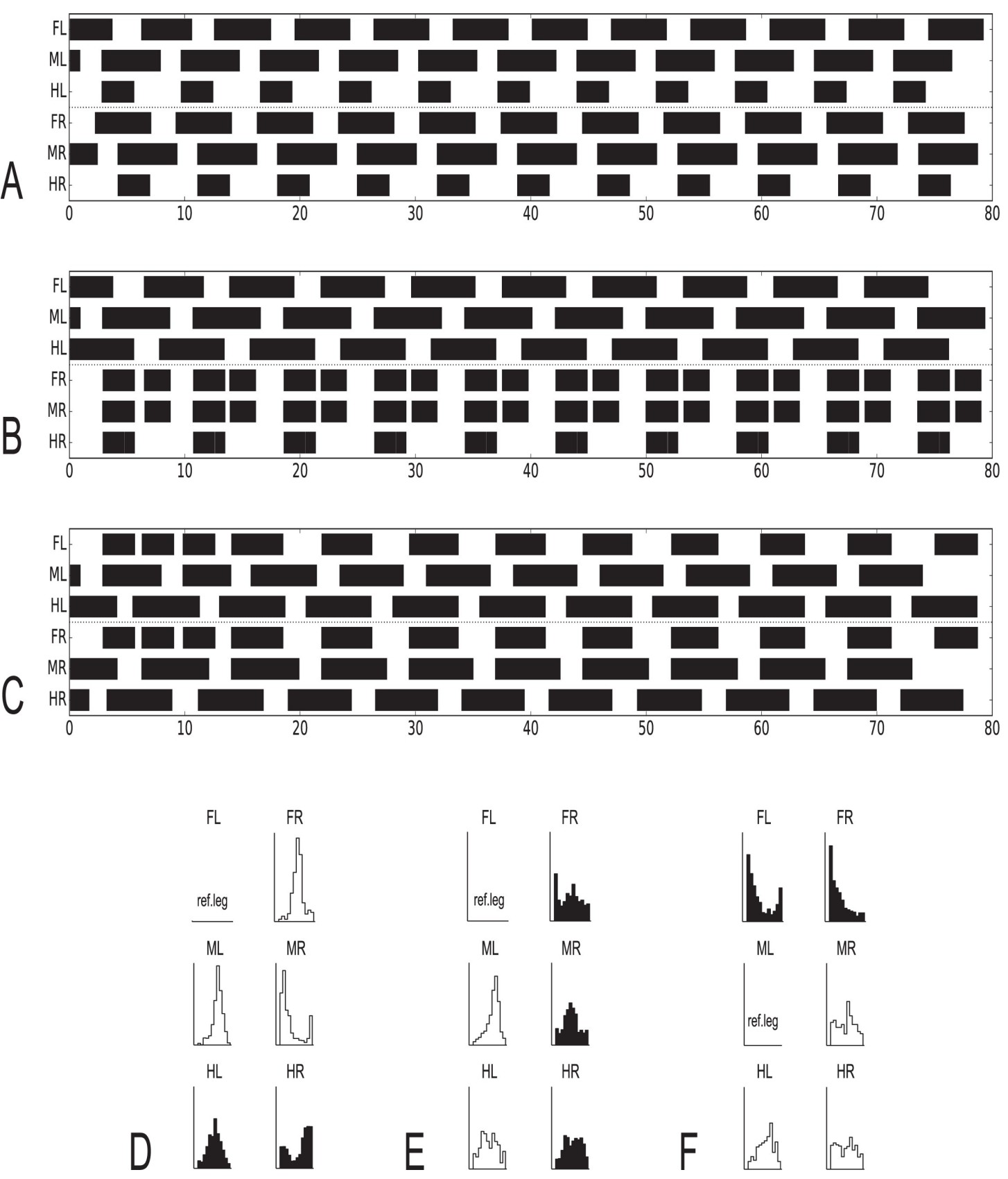

**Fig 10. Simulation of experiments where stick insects walk tethered on a treadmill with selected legs standing on force transducer platforms.** Shown are activations of retractor motor neurons (> 7 mV) over time (s) in A) to C). Corresponding experimental results [76,86] are given in D, E, F, respectively: Phase histograms of the beginning of the retraction in the walking legs (shown in white), force histograms of the maximum force in the standing legs (dark). Reference leg starting with the beginning of the retraction movement. In A) and D) both front legs and both middle legs are walking while both hind legs are standing. B) and E) shows left legs walking and right legs standing. C) and F) show both front legs standing, both middle legs and both hind legs walking [76,86]. Walking velocity neurons set to 30 mV.

Taken together, the simulations provide a good qualitative description of the experimental data, showing typical "gait" patterns, allowing for symmetry breaks and qualitative agreement concerning the ipsilateral delay depending on velocity. Further, contralateral phases of about 0.5 are observed for fast and for very slow movements and deviating values (around 0.7, 0.3) for intermediate velocities. Furthermore, there are stable patterns that might not be characterized as tripod or tetrapod. Thus, using this control architecture, adaptive walking emerges with different patterns only depending on a selected velocity.

## Backward walking

Whereas forward walking and negotiation of curves form a continuum with the turning radius between zero and infinity, backward walking represents a separate state as it is controlled by a separate motivation unit (Fig 2). The simplest solution we could find to introduce backward walking is to activate only rules 1 and 2i for ipsilateral coupling and use a version of rule 2c adapted to backward walking (using a changed threshold value), but deactivating rules 3i and 3c. Results for velocity neuron set to 50, 40, 30 or 20 mV are shown in Fig 7 (for videos see Supporting information S11–S14 Videos). Although the starting configuration is apparently not well suited for these tests, stable patterns are reached after a few steps. As for forward walking, patterns could be observed that might be classified as tripod (Fig 7A), tetrapod (Fig 7C), or stable patterns along the continuum, i.e. patterns that are neither tripod nor tetrapod, (Fig 7B) or (Fig 7D), or even between tetrapod and pentapod (which is not shown). These patterns resemble those observed in *Carausius morosus* [84] and are characterized by the ipsilateral sequence of swing movements hind–middle–front. This is different for results obtained from *Drosophila* [91] and for *Aretaon* [85], see Discussion).

## Analysis of neural activity–The possible function of Intrinsic Oscillation

To challenge the hypothesis mentioned above that CPGs are required to control walking, in the following we use our hexapod simulator to test if alternative interpretations may be possible that show the same emerging intrinsic oscillations.

**All legs deafferented.** We will first deal with the results of a study concerning experiments with locusts [69]. In all these experiments, the three thoracic ganglia of a locust have been deafferented. With this preparation, a number of experiments have been performed by treating only selected ganglia with pilocarpine. The main results are that all six hemiganglia oscillate with a period of about 5 s and show in-phase coupling (recordings from depressor motor neurons), if all three thoracic ganglia were treated with pilocarpine. This result is supported by studies on stick insects *Carausius morosus* [67,72]. Corresponding data are also available for rock lobster *Jasus lalandii* ([73]–recordings from retractor muscles, legs autotomized) and for crayfish *Pacifastacus leniusculus* [74]. When, however, in the study of Knebel et al. [69] only the metathoracic ganglion was treated with pilocarpine, right and left hemiganglia of all three thoracic ganglia showed anti-phase coupling. Similar results have been reported for *Carausius morosus* [54]. Knebel et al. [69] concluded that the oscillators of neighboring hemiganglia are all coupled via in-phase connections except the hemiganglia of the metathorax, which are coupled in anti-phase. Interestingly, the in-phase coupling observed corresponds quite well to

coordination rule 5 [15,92] as introduced into Walknet [86], see Fig 2, ocher units, 5i, 5c and Methods. In neuroWalknet the anti-phase coupling is realized as a connection between both metathoracic hemiganglia (rule 5ch, see Methods) coupled via inhibitory synapses (Fig 2, ocher units, upper left, lower left).

To what extent is it possible to simulate these results with our controller? To mimic deafferentation, we cut the velocity output to the motors thereby freezing all joint positions. Furthermore, the mode of each leg was set to stance and the inputs to the PMN units, marked by "pilo" in Fig 2 (and Methods) were set to high values, thereby simulating the effect of deafferentation and pilocarpine.

When, following the experiment of Knebel et al. [69], in our simulation either only the prothoracic ganglion or only the mesothoracic ganglion was "treated with pilocarpine", all six hemiganglia oscillated in-phase (Fig 8B and 8C, activation of depressor motor neuron). The same result was found when all three ganglia were treated with pilocarpine (Fig 8A). When however only the metathoracic ganglion was treated, ipsilateral hemiganglia remain in-phase, but all contralateral hemiganglia oscillated in anti-phase (Fig 8D). These simulation results agree well with the findings of Knebel et al. [69].

In the simulation given in Fig 8A, 8B and 8C, the antiphase influence occurring between the hind legs was apparently overridden by the stronger in-phase coupling of the other leg pairs. Note that influences from rules 1–3 were not effective in this situation as sensory input concerning leg position is fixed (and leg controllers were assumed to be in stance mode during the application of pilocarpine). Therefore, the critical effects resulted from rule 5 influences plus the extension assuming a contralateral inhibitory connection between the hind leg controllers (rule 5ch).

These simulations show that results of Knebel et al. [69] can be reproduced although the leg controller does not contain any CPG that triggers the rhythmic movement during normal walking, i.e. controls patterns characterized as pentapod, tetrapod or tripod. Rather, due to the pilocarpine driven high excitation of both premotor units of the beta joint and sufficiently high mutual inhibition via the red units (PMN) connecting both branches (Fig 2), the four-unit premotor network started to oscillate with a period of about 5 s.

As rule 5 connections are applied to the alpha joint, too, independent oscillations were also observed in the retractor-protractor system, which was however not (yet) investigated by Knebel et al. [69].

Coupling among both premotor units of the beta branches is assumed here in a way to produce levator and depressor activity with asymmetric ratio as observed in the animal (here we use a ratio of about 1: 2 duration). The connection weights could of course easily be changed to produce other, for example more symmetric relations.

**Intact walking legs and deafferented legs.**   In the following two sections, we show that neuroWalknet can even handle rather artificial situations that animals had been exposed to previously in experimental conditions. Borgmann et al. [68,72] studied *Carausius morosus*, where one intact leg, for example the right front leg, was walking on a treadmill while the other legs were amputated, i.e. deafferented only [68], or in addition treated with pilocarpine [72]. The authors recorded movement and retractor activation of the walking leg (right front leg) and neuronal activity of the other ipsilateral legs. Recordings from the retractor motor neurons of middle legs and hind legs being deafferented and treated with pilocarpine showed that all ipsilateral hemiganglia were driven in-phase with the stance mode, i.e. retractor activation of the walking leg. Interestingly, corresponding results have been observed by Clarac and Chasserat [73] studying rock lobster *Jasus lalandii* (legs autotomized).

How would neuroWalknet behave in this situation? Fig 9 shows the result when the left front leg (not the right one as used by Borgmann et al. [72]) was allowed to walk while all other

legs were deafferented and "treated with pilocarpine" as used for simulation of the Knebel et al. [69] experiments. In the simulation, all remaining ipsilateral hemiganglia oscillated in-phase and with the same period as the walking leg. This means that the oscillatory elements appear to be driven by the rhythm of the walking leg. As shown in Fig 9, the simulation further predicts that contralateral legs should oscillate in-phase. To our knowledge such recordings have not been performed yet. Simulation of different walking velocities are provided in Supporting information S4 Fig indicating that the right hind leg appears to be less strongly coupled to the walking leg than all other legs. Note that this simulation may also be considered as an explanation of results of Wendler [93], who observed stumps of amputated middle legs moving in-phase with the intact ipsilateral front leg walking on a treadwheel.

**Standing legs in walking insect.** Finally, a set of experiments will be discussed, where animals were supported above a treadmill, but only a few legs were able to walk, i.e. able to move the wheel. Different to the case of Borgmann et al. [72], the other legs were not deafferented, but remained intact and were being placed to stand on a force transducer each being fixed aside the body. These standing legs often performed oscillating motor output [94].

A preliminary experiment of this kind has been performed with the rock lobster (*Jasus lalandii*), but with only one leg on a force transducer [95]. The leg standing on the fixed force transducer showed rhythmic, backward directed forces in-phase with stance movement of the anterior neighboring walking leg. More detailed experiments were performed with stick insects *Carausius morosus*, where different configurations of walking and standing legs have been studied [76]. It was however difficult to obtain clear results in some of these experiments, probably because in these quite unnatural situations rules 1–3 influences and rule 5 influences are superimposed leading to irregular behaviors difficult to analyze. Another problem concerns simulation of these experiments. Animals are supported above a treadwheel. However, for technical reasons a robot could not be supports above a treadwheel. Instead, the simulated robot is walking with three or four legs, while the fixed legs are not allowed to move but are therefore slipping on the ground which may lead to not well structured walking movements. We therefore focus on only those three cases where at least three neighboring legs are walking. In these cases, a separation between rule 1–3 effects and rule 5 effects appears to be possible at least to some extent. Here we show results and simulation of those cases where the experimental results showed obvious maxima concerning appearance of posterior extreme positions (PEP) of walking legs and force maxima of standing legs.

First, we consider the case, where both front legs and both middle legs that walk on the treadmill show an about normal walking coordination (Fig 10A, simulation; D, biological data). Each of the standing hind legs showed strong forces pointing rearward in-phase with the retraction of the ipsilateral middle leg. This result is recognizable in both behavior and simulation. How could this result be explained? In this situation, rule 3i-influences from middle legs to hind legs were not effective as the hind legs adopted a constant position in stance mode. Rule 5-influences from hind legs to walking middle legs had no effect when the middle leg was in swing mode. If the middle leg was in stance state, there was no specific change in leg velocity as all legs walking are mechanically coupled via the substrate. However, the standing hind legs appear to have received rule 5-influence from their neighboring middle legs elicited due to high friction in the middle legs walking on the treadmill. This effect was stabilized further by the anti-phase influence of rule 5ch active between both hind legs. Thus, there is an at least qualitative agreement between behavior and simulation.

Another interesting case (Fig 10B) is given when the three left legs were walking on the wheel while each right leg was standing on its force transducer. The walking left legs showed a pattern somewhat slower than ideal tripod (behavioral data, see Fig 10E). The standing right front leg showed two force maxima, one in-phase with beginning of stance of the left, walking

front leg, the second one being in-phase with force maximum of the right middle leg. The latter was in-phase with stance beginning of the left, walking middle leg (an in-phase influence to the right middle leg from the first maximum of the right, ipsilateral, front leg may exist but is not significant).

How could this behavior be explained? The right front leg received rule 5-influence from the left front leg when starting stance which lead to its first (left) maximum. The second maximum was elicited by rule 5 input from the ipsilateral, standing middle leg, which in turn resulted from the beginning of stance of the left, walking middle leg. An ipsilateral rule 5-influence from the right front leg may exist, but if so, might have been suppressed by less focused rule 5 influences from the standing (right) hind leg. The bimodal force output of front leg and middle leg is clearly visible in the simulation. The unclear force distribution in the right hind leg may result from opposing influences, first the ipsilateral rule 5 influence from the right middle leg and second the weaker anti-phase influence via rule 5ch from the left hind leg. There is a broad maximum in the standing hind leg (Fig 10E), but it is not clear if this maximum agrees with the one observed in the simulation.

The interpretation of the case shown in Fig 10C and 10F–both front legs are standing on a force transducer–seems more difficult, as there is only weak coupling between right and left walking legs. This may result from lacking of 3i-influences from front legs to middle legs which includes lacking 2c and 3c coupling via the front legs. A phase shift between contralateral neighboring walking (middle and hind) legs is also observed in the simulation. In contrast, both standing front legs showed strong in-phase coupling, being recognized in behavior and simulation (Fig 10C and 10F). Extreme in-phase coupling between front legs had been reported by Cruse and Saxler ([94], their Fig 3) suggesting a quite strong rule 5-influence, at least in a situation where both front legs try to pull the body forward. The asymmetric distribution of the force maxima in the front legs results from the fact that left middle leg was used as reference in this evaluation.

Taken together, the behavior observed in quite unnatural situations of standing legs in a walking animal appears to be, at least in principle, in accordance with the properties implemented in neuroWalknet.

## Discussion

How to control natural behavior that requires more than just controlling simple reflexes is still a question under debate. 'Natural behavior' means that the agent has to deal with a complex, unpredictable environment as well as with a motor system that is characterized by a high number of degrees of freedom (e.g., joints). These DoFs may be arranged serially or in parallel and usually include redundant DoFs, i.e. an arrangement where several solutions are possible to reach the same behavioral goal. Hexapod walking and climbing on unstructured substrate represents such a case. Here, we propose an artificial neural network, neuroWalknet, a decentralized controller which is suited to approach such problems and which represents a testable hypothesis as to how walking behavior as studied in insects may be controlled. The neuronal basis of the neuroWalknet approach complements the original Walknet approach [23] with a detailed neuronal realization based on an antagonistic structure.

This neural-based control approach was tested in different behavioral settings. neuroWalknet is able to produce footfall patterns that include "tripod" patterns, "tetrapod" patterns, "pentapod" patterns as well as various stable intermediate patterns. These patterns form a continuous multitude as has been observed in stick insects [34] and in *Drosophila* [49], which has recently been supported by a detailed study of DeAngelis et al. [50], and to some extent in cockroaches (e.g., [13,48]). The patterns are robust with respect to disturbances provided by

different starting configurations, for example. For a specific range of velocities, mirror image patterns are possible with phase coupling among contralateral legs deviating from phase values of 0.5. Patterns recently described as non-canonical by DeAngelis et al. [50] can be observed, too (e.g., Fig 3C). Although for *Drosophila* DeAngelis et al. [50] report phase values of 0.5 for contralateral legs, their individual examples show deviations from antiphase as found in our simulation. The phase value of 0.5 might therefore result from averaging possibly not recognized mirror image walks. Furthermore, neuroWalknet allows for backward walking and negotiation of curves. A leg controller of this type may be classified as a "free-gait" controller [23] because the network does not rely on motor memories representing explicitly defined specific patterns. Rather, the behavioral patterns observed reveal emergent properties of a decentralized neuronal architecture, which easily allows for starting or interrupting a walk or changing the velocity. Thus, the new controller corresponds well to established behavioral data.

In addition, the controller allows to simulate results of neurophysiological studies generally assumed to support the concept that CPGs are required to control the rhythmic movements of walking legs. We show that control of walking can be explained without relying on this concept.

In general, neuroWalknet is a dynamical systems approach from which activities emerge in interaction between different decentralized components of the control system and with the environment mediated through the body. In the following, we will compare neuroWalknet with corresponding biological data on the neuronal level, the level of the leg controller and that concerning interleg coordination.

## A) Neuronal counterparts

neuroWalknet may provide a functional counterpart for some neurons described in neurophysiological studies, thereby providing an idea as to how these neurons might be embedded into a complete system.

*Representation of Walking Parameters on the neuronal level*: In *Drosophila*, Bidaye et al. [91] and DeAngelis et al. [50] found and characterized two 'moonwalker descending neurons' (MDN) on either side in the protocerebrum, which may stabilize backward walking. These neurons may functionally correspond to one of the four motivation units required to decide between forward mode and backward mode and to stabilize the latter (Fig 2, rightmost dark gray units, "backward") activated by antennal stimulation. Furthermore, they found a 'moonwalker ascending neuron' (MAN). This unit may correspond to the upper central dark gray unit in Fig 2, which stabilizes backward walking by inhibiting the leftmost dark gray unit ("forward").

Kai and Okada [96] described a group of neurons observed in crickets that fire closely related to walking velocity. Functionally related neurons have been found in the central complex of cockroaches [9]. These neurons may correspond to the neuron shown in Fig 2 (leftmost white unit) as it represents the current walking velocity.

*Neural representation of Searching Movements as part of an Adaptive Swing Movement*: In stick insects, searching movements of a leg are observed if the leg at the end of the swing movement is not stopped by ground contact [35]. Frequencies of searching movements are about 3 Hz (and about 10 Hz in *Drosophila*, [97]). In both cases, these values correspond to that of the uppermost leg frequency during fast walking. One may therefore ask if such searching movements represent a distinct behavior which would be addressed separately on the level of motor selection, or if searching movements just simply are part of an adaptive swing behavior. In an excellent study concerning the control of searching movements, Berg et al. [98] investigated

the contribution of non-spiking inhibitory interneurons during stance and during swing as well as during performance of searching movements in a restrained middle leg and found two antagonistically active groups. One group is hyperpolarized during flexor activity, while neurons of the other group are depolarized during flexor activity. Two neurons, I1 and I4, each belonging to one of either group, have been studied in more detail. I1 belongs to the former group and is active during stance, I4 belongs to the latter one and is active during swing movement and during searching movements in *Carausius morosus*. Berg et al. [98] could show that swing and search is stopped either by foot contact or by hyperpolarization of I4. Artificial neurons of this type are found in neuroWalknet, too, whereby the lower light grey unit in Fig 2 may correspond to a neuron like I4 and the upper light grey unit may correspond to a neuron like I1. If this interpretation is correct, an additional neuronal unit representing a searching movement would not be necessary, contrary to what has been postulated by Berg et al. [98] and tested in a simulation by Szczecinski and Quinn [99]. Therefore, swing and search behavior may result as an emergent property from the network already required for basic walking. Consequently, the oscillations observed during searching result from the properties of negative feedback loops controlling the joints during swing as given in neuroWalknet (see also [35]).

*Ring network and Central Complex*: A hypothetical ring net as proposed here to exist in each thoracic hemiganglion (Methods, Fig 11) shows parallels to the ring network forming the central complex, a prominent structure in the insect brain. As interpreted by Stone et al. [100] and others, this structure may serve for vector navigation. To this end, it receives input used to record the direction of current body axis in a geocentric coordinate system and information concerning the currently desired walking direction theta measured in a body-fixed coordinate system. These signals are projected to the noduli. Stone et al. [100] postulate that the noduli provide a vector representing the walking direction which is given to the motor system (for neurophysiological data see also [101]). Our proposed ring net (Methods, Fig 11) receives as input angle theta and the current leg position represented by angle alpha, and provides as output a direct influence on alpha angle and on gamma angle of the corresponding leg, as described in Methods, representing the components forming the leg trajectory. This means, when compared with the central complex, the leg ring network might represent a similar, but much simpler structure where layers 2 and 3 correspond to the input to the protocerebral bridge and layer 4 and 5 plus units O1 –O4 (Fig 11) represent the functional aspect of the noduli. Neurons providing information from the brain to the hypothetical ring net of each leg may be represented by descending neurons as for example described by Träger and Homberg [102].

One may therefore speculate that the ring net, assumed to be realized in each thoracic hemiganglion, may be an evolutionary early, simple network suited for controlling the individual legs. The central complex may have arisen later as a merged version of right and left ring networks thereby forming a centralized and more sophisticated structure required for controlling the walking direction of the animal to be used for vector navigation [100,103]. Information concerning desired walking direction provided by the central complex may then be given to each of the six ring nets.

## B) Individual leg control

neuroWalknet is based on detailed neural control networks and includes antagonistic structures. This allows to analyze and compare different hypotheses on the neuronal level.

*Control of the movement of an individual leg*: For their model, Daun-Gruhn and Büschges ([26], their Fig 3) introduced an interesting distinction between "timing" and "magnitude". Timing is controlled by a structure based on lateral inhibition, in their case a CPG that

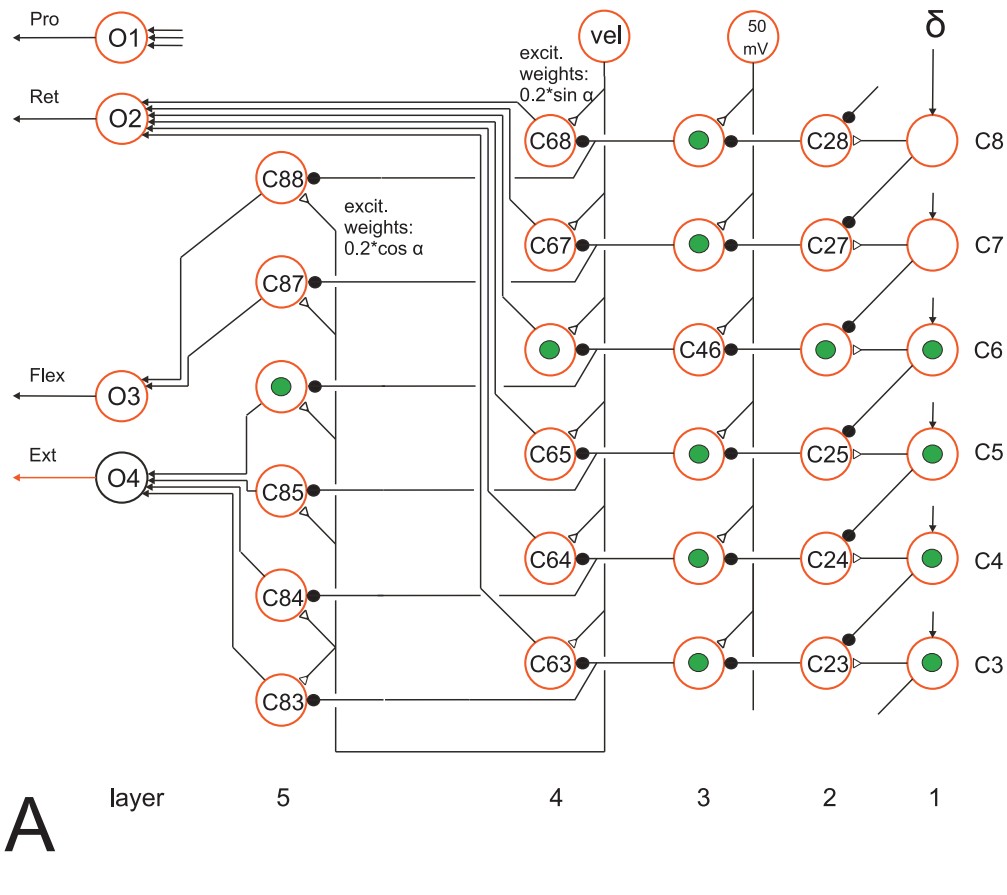

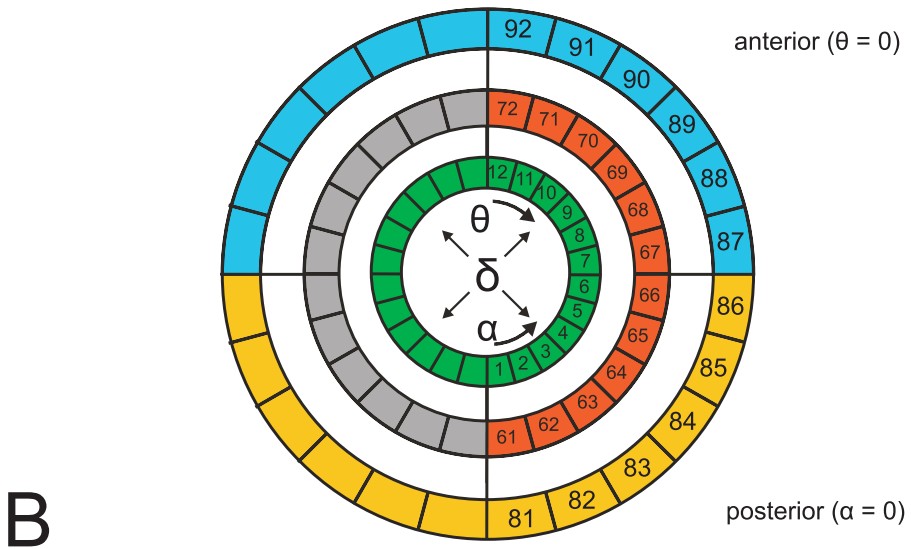

**Fig 11. Ring net.** A) Angle delta (leg position, angle alpha, and walking direction, theta) determines the contribution of alpha joint and gamma joint during stance. Column 1 (right) represents angle delta in spatial coding. Depending on desired velocity of the leg (vel) and angle delta the motor output for the retractor (O2), for flexor (O3) and for extensor (O4) are computed. Protraction (O1) is not yet implemented. Units marked with green dots show, as an example case, activation of the net by a delta value input of spatial code 6 (e.g., theta = 0 degrees and alpha = 25 mV). B) Organisation of ring net, schematically. Input: layers 1–3, green. Input delta = alpha–theta. The analog value delta (in mV) is transformed to the spatial coding version (Fig 2, box "spatial coding"), which is then given to the circle marked green. Output: Layer 4 (red, grey) controls the alpha joint (red: retractor, unit O2, green: protractor, unit O1, not used in the

current version). Layer 5 (blue, yellow) controls the gamma joint (blue: flexor, unit O3, yellow: extensor, unit O4). Letter C (for numbers of circle units, Fig 11A) is not shown in Fig 11B.

depends on internal commands (e.g., global velocity) and on sensory signals. Magnitude concerns the determination of the strength of motor output depending on sensory feedback. In principle, this concept is also applicable to the structure of neuroWalknet. The main difference to neuroWalknet is twofold. First, there is only one controller for timing, represented by the four light grey units, that concerns all three joints of a leg, whereas in the model of Daun-Gruhn and Büschges [26] and of Rubeo et al. [77] each joint has its own "timer", the CPG of the respective joint. Second, in neuroWalknet the "timer" is realized as a bistable monopole, a network that can adopt one of two possible stable states. A change from one state to the other is only possible if strong enough external signals are provided. As shown in Fig 2, the leftmost ("swing") and the rightmost ("stance") light grey units receive input from sensors (leg position, load) of the own leg and, via coordination signals, from neighboring legs, which in turn depend on the positions and on the state of units "swing" and "stance" of those legs as well as on the global velocity signal. Note that this "timer" does not operate with time explicitly, but instead exploits embodied time signals.

A bistable monopole, given by mutual inhibition between units "swing" and "stance" is indeed necessary for two reasons: When the hexapod starts with an uncomfortable leg configuration, both units might receive contradictory signals which requires a decision as to which of both units should be activated. But the inhibitory connections are also necessary during stable walking, because during walking there are time periods where none of both units receives input. In this case, the mutual inhibition system acts as a short term memory maintaining the actual state until new input is arriving. Application of a bistable monopole is in full agreement with the contributions of Bässler [18] and Pearson [22] concerning a thorough definition of CPGs. Correspondingly, the network controlling states forward and backward (Fig 2, dark grey units, upper right), another bistable monopole, does not oscillate, either.

For simulation of cockroach running, Rubeo et al. [77] use CPGs for each joint as a basis to control walking rhythms. They connect neighboring legs by direct coupling of the flexor muscles MNs (corresponding to the levator muscle of stick insects) through coordination rules 1–3. Different to Hector, the authors applied a much more sophisticated simulation of, in this case Hill-type, muscles focusing on tripod pattern with a given velocity. The most important difference is that in neuroWalknet, leg control depends on the swing–stance pair of units (marked by light grey units). As a consequence, in neuroWalknet control of joints is not strictly connected to the higher-level control represented by the light grey units. Therefore, different controller types can easily be used for either swing or stance. But also within stance state or swing state variable application of antagonistic muscles is possible (e.g., S1A Fig, e.g., alpha joint, gamma joint). This means that implementation of avoidance reflexes or short steps as well as searching behavior appears easy within neuroWalknet, because a local change of antagonistic muscles does not influence the overall state of the leg. Furthermore, as no CPGs are applied on this level, our structure strongly simplifies height control, negotiation of curves and dealing with disturbances, as no dynamical oscillatory systems have to be constrained. This is particularly obvious for starting and stopping. As shown by Tóth and Daun [87], in their system based on coupled oscillators various conditions have to be fulfilled to overcome the dynamic behavior of such a system (related problems are described by Rubeo et al. [77]). This is in strong contrast to our solution, where simple setting of global velocity from zero to a positive value or back is sufficient. This means that application of the structure representing a bistable monopole increases the adaptivity of the behavior dramatically and shows that for

walking CPGs are not necessary, but instead may rather cause negative effects. Different approaches on legged robots apply simple CPG approaches for temporal coordination [27,104]. While these excel at spatial coordination in rough terrain, recently Bellicoso et al. [105] argued that—for their case of a quadruped robot—climbing and changing environmental situations also require adaptation on the temporal level which appeared difficult when dealing with fixed gaits. Operating with CPGs the approach of Rubeo et al. [77] resembles more our simulation of running (Supporting information S2 PDF and in S9 Video).

Akay et al. [106] show that, during active walking, stimulation of specific campaniform sensilla (CS) elicit a switch from protractor activation to retractor activation, which might be interpreted as to switch from swing to stance. Interestingly, the authors show that such an influence can also be observed when the insect is deafferented (except the CS input) and treated with pilocarpine. They also show that the CS influence does not influence the motor neurons directly, i.e. does not circumvent the system that is responsible for the oscillations elicited by pilocarpine. As a next step the authors assume that the network showing artificially elicited oscillations are exactly those structures that are responsible for controlling the walking rhythm assuming a minimal structure (Ockham's razor). From our point of view, this introduces the problem that they take this as a basic assumption without considering other possibilities or explanations. Here, we want to point out the difference to our approach: We do not take this assumption as the given cause of any rhythmic locomotion. Instead, we assume that control of walking "rhythm" is located at a higher level, the network shown in light grey in Fig 2. That means we have two networks, one, the PMN net, that oscillate only when treated with pilocarpine, and another one, the light grey net, that allows for quasi-rhythmic motor output, which does, however, not operate as a central pattern oscillator. Rather, it is part of—as it has sometimes been called—, a peripheral oscillator (i.e., based on sensory input).Therefore we propose as an alternative that, for walking, no CPGs are used but only one bistable monopol per leg, which has advantages on the functional level as mentioned above.

*Levator Reflex*: The levator reflex in *Carausius morosus* is elicited if during swing movement the leg touches an obstacle. In this case, the leg shows a brief retraction accompanied by a levation, followed by protraction to continue the swing movement (e.g., [78,107]). This reflex has been implemented as it can easily be simulated if a tactile stimulus to femur or tibia is represented by a fictive sensor and the stimulus is given via a neuron to the retractor branch and to the levator branch as has been shown by Schumm and Cruse [78].

## C) Interleg coordination

In the following, we will further discuss specific aspects concerning interleg coordination in more detail.

*No Central Rhythms required for slow walking*: neuroWalknet, a free gait controller, enables not only the emergence of a continuum of gaits as found in insects, but can also produce central rhythms as observed in a number of neurophysiological studies on insects [54,67,69] and crustaceans (*Jasus lalandii [73]*; crayfish [74]). Furthermore, several experiments can, on a qualitative level, be explained in which either neurophysiological recordings in deafferented legs are combined with legs walking on a treadmill [72] or where intact legs standing on a fixed force transducer are combined with legs walking on a treadmill [76]. In the latter case, oscillatory motor output can be observed although no pilocarpine has been applied. Interestingly, in principle corresponding results have also been observed in crustaceans [73,95], arguing for a general relevance of a controller structure proposed challenging the generally accepted view that CPGs observed in different insects or crustaceans do represent the basis for the control of multi-legged walking.

How can footfall patterns as tripod, tetrapod, pentapod or intermediate patterns emerge depending on velocity only although neuroWalknet, for forward walking, does not contain separate subnets to be switched on or off, nor are there neuronal weights being varied? To provide a qualitative explanation, we introduce four "dummies" D1 –D4 that symbolize seemingly in-phase coordination influences between diagonally neighboring legs as indicated in Fig 4B. In neuroWalknet these four diagonally active dummies are based on coordination rules R2i and R2c. Specifically, D1 acts between HL and MR (realized by R2i from HR to MR and by R2c from HR to HL), and correspondingly D2 between ML and FR, D3 between HR and ML, and D4 between MR and FL.

In short, the effect of these four dummies can be characterized as follows. For velocity neurons set to values about equal or larger than 40 mV, all four dummies operate together supporting each other and thereby stabilize a tripod pattern. As velocity input decreases below 40 mV, such a cooperation is not any more possible. In the range from 32.5 mV until 15 mV only two of the four dummies (either D1 and D2 or D3 and D4) cooperate effectively by supporting each others swing movements. In this range, the attractor allows for tetrapod patterns with the corresponding diagonal leg pairs showing quite strict synchronies. The other two dummies are not effective because their signals appear in an early leg position where swing cannot be elicited yet. They are however able to activate the corresponding mirror image version (e.g., Fig 4B). The power of the dummies decreases with velocity due to the high pass filter properties of R2i (Fig 2). Therefore, they are not any more effective below velocity inputs of 15 mV, which leads, for velocity inputs equal or smaller than about 10 mV, to pentapod patterns and phase values of 0.5 between contralateral legs. As shown in S2 Fig, in the cases of 37.5 mV and 35 mV stable intermediate patterns can be observed. Similarly, stable patterns can be observed around velocity neurons set to 12.5 mV which however are neither tetrapod nor pentapod.

*Testing other types of coordination between legs*: Concerning coordination influences between legs, Tóth and Daun [87] postulate that various direct coupling connections between hind legs and front legs are necessary to establish coordinated locomotor pattern. There are, however, no experimental hints supporting such an assumption. Their conclusion appears to be valid only under the assumption that CPGs are indeed responsible for control of walking. neuroWalknet does neither contain direct coupling between front legs and hind legs, nor does it use CPGs to control walking rhythm, but nonetheless allows the control of different patterns plus stable intermediate patterns as observed in stick insects and *Drosophila*. Thus, neuroWalknet can explain more data with less effort.

*Interleg coupling dependent on walking velocity*: Along a similar line, Borgmann and Büschges [10] report that coupling is weaker in slow walking insects compared to fast walking. Indeed, the simulation with neuroWalknet shows that for very slow velocities (e.g., velocity neurons to 10 mV or less) and arbitrary starting configurations it may take more time until a stable pattern is reached. However, weak coupling observed on the behavioral level does not mean that neuronal parameters, i.e. synaptic weights, are changing depending on velocity. In neuroWalknet this effect is simply caused by signals that–sent via the coordination channels– occur more rarely in slow walking. Therefore, more time is required until a stable pattern is reached after a given disturbance.

DeAngelis et al. [50] propose a very simple network for interleg coordination in *Drosophila* with contralateral legs using only phase value of 0.5, ipsilateral legs using only forward directed influences and all parameters are independent of walking velocity. For stick insect (*Carausius morosus*) detailed studies show that ipsilateral coupling operates in both directions and contralateral legs show phase values that differ from 0.5 during tetrapod walking. These data and those from Graham [34] further show that both ipsilateral coupling and contralateral coupling depends on velocity. Therefore, the coordination influences in *Drosophila* may be simpler

than those found in *Carausius*, or studies that reveal more detailed behavior as performed with *Carausius* (review [14,23]) are lacking in *Drosophila*.

*Interleg coupling dependent on walking direction*: As illustrated in Fig 7, walking backward shows stable footfall patterns similar to those observed during forward walking. This seems to be in contrast to the results of Pfeffer et al. [108], who stated that coordination in backward walking ants is weaker than in forward walking. The reason for this weak coordination may be that in these experiments ants had to carry higher load when walking backwards which provides irregular sensory input that may disturb the controller or, that the ants simply adopt a lower velocity (see above). So, actual coordination observed on the behavioral level is weaker indeed, but this does not necessarily mean that neuronal parameters responsible for interleg coordination have changed.

*Backward walking*: Szczecinski et al. [109] could show that a more statically stable version of hexapod backward walking can be observed when the direction of the ipsilateral sequence "rear-middle-front" of legs swinging is reversed to "front-middle-rear". This sequence has indeed been observed in *Drosophila* [91] and in *Aretaon asperrimus* [85], but not in *Carausius morosus* [84]. Here we adopted the simple version that leads to the *Carausius* pattern. The alternative pattern, which may meet the situation of other insects, would at least require introduction of additional, inverted versions of rules 1 and 2i, that operate from any leg to its posterior neighboring leg. Qualitative behavioral observations indicate that *Carausius morosus* appears to avoid spontaneous backward walking, which suggests that this species may not be equipped with further specific rules allowing for more "comfortable" backward walking, in contrast to species like *Aretaon* or *Drosophila*.

*Curve Walking*: One well documented case of curve walking [57] allowed us to test the ability of the controller to negotiate curves (Fig 6). We have introduced a new neuronal element, "local (theta)" (Fig 2, box local (theta)), that, together with the ring network, is able to allow for negotiation of curves. However, this was only done as a proof of concept for that specific turning angle theta, because only for this case sufficient data were available. The results are similar to the data provided by Dürr and Ebeling [57], but there are also obvious differences. Specifically, there are more irregular steps of the inner hind leg observed in the animals compared to those seen in simulation. This may be due to swing movements not being recognized in the experiment because the vertical amplitude is small and the leg may not leave the ground due to the current load distribution of the specific leg configuration. Therefore, simulation results recording neuronal activity (Fig 6) could appear as more regular than the behavioral results ([57], their Fig 2Aii).

Taken together, simulation of this specific case shows sufficient agreement with the behavioral results and might therefore be considered as a proof of concept for control of curve walking. However, more biological data would be required to expand the simulation for being able to describe turning on a more general level.

*Stability of behavioral patterns*: Studying the stability of behavioral patterns, as in our case represented by the footfall patterns, is difficult as the system may be characterized by showing bifurcations where small disturbances may lead to large effects. The system might reach its attractor via a short trajectory or it might require a longer trajectory, or, if there are two attractors (see example Fig 4), it might switch to the other attractor. Apart from choosing specific starting configurations we have performed systematic tests with different legs, different velocities and different durations of disturbances. Detailed results are, however, not shown here, as only general qualitative observations are possible (but see specific examples in Figs 3 and 4). In all cases studied the disturbance only affects a couple of steps after which a stable pattern was reached again. Further, in general, as to be expected, lower walking velocity leads to smaller deviations and small disturbances lead to comparatively small deviations from the stable

pattern. Overall, these observations are in full agreement with the many observations and different analyses of patterns found in the former Walknet approach ([23] for a review).

*Running*: Cockroaches, one of the most intensively studied insect as concerns walking, are often characterized as typical tripod (or "double tripod") runners. However, as already reported by Hughes [48] and [38] they also use non-tripod gaits. This is supported by results of Bender et al. [13], Full and Tu [19,110], Ting et al. [111] who found, in cockroaches, a continuum of walking velocities from zero to 0.5 m/s, corresponding to a frequency of nearly 15 steps/s. Therefore, as mentioned earlier for running a controller may be installed that operates without sensory feedback. The generally proposed solution to this problem is to apply CPGs, which may possibly indicate a further evolutionary development allowing for very fast walking or running.

In the Supporting information we show how a corresponding expansion can indeed be implemented in neuroWalknet (Supporting information S2 PDF). Importantly, this expansion exploits (i) already existing structures to introduce CPGs for controlling rhythmic movement, (ii) an intraleg coupling inspired by results of Büschges et al. [67], and (iii) an interleg coupling based on results of Pearson and Iles [112]. As for all open-loop control approaches this requires a form of stabilization to counter drift. In CPG-based approaches this is often realized through introducing mechanical limits (which may include nonlinear properties of muscles) on joints or some modulating sensory feedback [113]. This allows for stable walking controlled by cyclic motor output in our case as well. High-speed bipedal running observed in cockroaches [19] may represent an evolutionary later state, which is however not considered here.

To summarize, many behavioral studies and simulation models take central rhythms as a basis of hexapod walking and see CPGs playing a causal role. But this concept is based on mostly neurophysiological studies and on an only very limited number of behavioral studies under usually quite restricted conditions. For more natural behaviors that require a much higher adaptivity and are characterized by a much higher variability, such approaches seem ill-suited as they assume that the role of sensory information gathered during locomotion only plays a subordinate role. Indeed, such a secondary role of sensory signals required to overcome sensory delay appears an important feature for running in contrast to hexapod walking. Such a distinction of heterogeneous contributions of sensory information on motor control depending on behavioral state might as well be present in mammals (for example, in a recent study, Clancy et al. [114] found differing connection strength from sensory areas to motor control areas depending on behavioral state and related to locomotion. The authors argued that locomotion might shift cortical control networks more and more into a regime giving greater relevance to sensory feedback for slower velocities.). Adaptive locomotion as found in slow walking, negotiating curves and climbing through twigs seems to require a lot more sensory information for stable behavior. Therefore, neuroWalknet as proposed here agrees well with these findings. It furthermore agrees with principles for hierarchical motor control as proposed by Merel et al. [115] for mammals. These, partly overlapping, principles concern (a) *information factorization* as different sensory signals are routed to different sub-sections of the network, for example at lower-level control for swing and stance and at higher-level control for forward walking or backward walking, (b) *partial autonomy* as, for example, local reactions to disturbances (e.g., Levator reflex) for lower-level elements, (c) *modular objectives*, as the separation between swing control system (Fig 2, green and red units) and stance control system (Fig 2, blue and red units), (d) *multi-joint coordination* as represented by the swing system controlling swing trajectory, (e) *temporal abstraction* simplifying the control task as higher-level tasks (walking forward vs. walking backward) are separated from lower-level tasks as control of the individual leg, both operating on different time scales, (f) *lower-level movement*

*centers* that generate their own spatiotemporal coordination patterns (e.g., of a leg) which may be realized by structures with or without CPGs.

## Conclusion

In this paper, we suggest a new view on the structure of the motor control system of insects. The overall behavior emerges from the interaction between various decentralized 'motor memory' elements organized on different levels and the environment mediated through the body. This view is supported as neuroWalknet can reproduce a wide variety of behaviors and allows in addition to simulate a number of experimental data concerning specific neurophysiological studies, never reached by a former approach.

This is possible because the sensory-motor memory elements are connected in a variable way in the motivation unit network, a recurrent neural network allowing for forming attractor states. Units of this net can be found on several levels, the leg level (swing, stance) or higher levels (forward, backward, run). This structure enables the implementation of even higher levels as are, for example, states like "Searching for food" or "Homing" [24,116]. Symbolic elements may not be relevant for a neural structure of stick insects, but there are strong arguments that such elements may be required in other insects, for example honey bees or bumble bees [117,118].

neuroWalknet adopts a view different from generally proposed assumptions because the contribution of CPGs is interpreted in a fundamentally different way. Following our interpretation the basic controller concerns slow walking, with strong impact on walking on irregular substrate including the requirement of developing higher forces in longitudinal direction. These properties are realized in both insects and crustaceans, whereas CPG-based running is assumed to having been developed only later. We believe that this view might stimulate comparative studies going beyond the traditional concentration on a small number of species only.

## Methods

The simulator consists of two parts, one for the neuronal controller and one for the body of the hexapod robot Hector, which exists as a hardware version and as a dynamic simulation (implementations are publicly available: dynamical simulation environment is realized in C++ and based on the Open Dynamics Engine library, see https://github.com/malteschilling/hector; the neuroWalknet controller has been implemented in python (version 3), see https://github.com/hcruse/neuro_walknet). Here we use the dynamic simulation. In the following, we will, first, describe the simulation environment, and second, the realization of the neural controller including the neuron model. Finally, we will give details about how specific experiments were realized using the simulation.

### The simulated robot Hector

The robot Hector contains six legs with three active joints each and is designed to reflect important aspects of the geometry of stick insects with a scaling factor of 1:20 (Fig 1). The three joints are called alpha joint for the Thorax-Coxa joint, beta joint for the Coxa-Trochantero-femur joint, and gamma joint for the Femur-Tibia joint. Each joint is constructed as a hinge joint (which is a simplification, because in stick insects the basic joint has more than one degree of freedom, and in other insects additional joints can be found). The joint rotation axes are oriented as is the case in stick insect *Carausius morosus*, which means that the axes are not orthogonal, but show different inclinations for each leg [81]. Mechanoreceptors recording joint position and higher derivatives exist in each joint. In insect legs hair plates or hair rows are found in the Thorax-Coxa joint and the Coxa-Trochanter joint. The prominent sensor in

the gamma joint is the femoral Chordotonal organ. However, in all joints there are various other sensors, but it is not known in detail in which way they contribute to the motor control system. Therefore, as a further simplification, we assume that, for each joint, there are antagonistically structured position sensors [76,78,79]. This means that each joint is assumed to contain two sensor structures each of which monitors the full range of joint position, but with increasing excitation in opposite directions, i.e. we deal with redundant structures.

For simplicity, the three leg segments (Coxa, Trochanterofemur, Tibia) are identical for all legs. This differs from the insect where in front legs and in hind legs the Tibia (and to a smaller extent the Femur) is longer than that of the middle leg. Due to the size of the motors and the fact that all legs show the same geometry, the joints have a smaller range of movement compared to that of the animals. Therefore the maximum size of an average step used here is about 33 cm compared to the 40 mm in stick insects, defined by average values of anterior extreme position (AEP) and posterior extreme position (PEP) as used during normal, undisturbed walking. Different to the animals, there is no tarsus, and therefore there are no adhesive structures at the end of the foot. Another difference concerns the position of the center of mass (COM), which in the stick insect is placed between the Coxae of both hind legs, whereas in Hector it is placed slightly in front of the Coxae of both middle legs [119].

**Measuring the position of the joint and estimating the torque.** The simulator and the robot are equipped with joint position sensors that are used for a negative feedback position controller. Neither the robot nor the simulator does (yet) contain explicit force sensors. However, loading or unloading a leg may provide important information. As there are no explicit load sensors given in robot Hector, we recorded the desired position of each joint and the actual position reached after the elastic element of the motor. The difference is used as an approximation of the torque produced by the motor (see S1B Fig).

**Joint actuation.** The purpose of joint actuation is to enable leg trajectories typical for a walking insect. To this end, motorneuron activities drive joint movements via muscles. For a specific muscle, the extensor tibiae muscle of stick insects, a large body of complementary research exists based on physiological studies and simulations [120–128]. In these experiments, legs were cut except one fixed middle leg. The extensor muscle of this leg was then stimulated with various artificial signals. Results showed that effects of inertia and momentum are small in contrast to animals with large limbs. Continuing their earlier studies, v. Twickel et al. [127] recently proposed a sophisticated Hill-type muscle model that also included a passive antagonist, i.e. flexor muscle. The authors could show that, using this model, swing movement of the gamma joint could indeed successfully be simulated.

As for the robot, different to the insect, inertia and momentum are relevant, we had to take another and more abstracted approach that allows for producing leg trajectories similar to those observed in the stick insects. Inspired by results of Mu and Ritzmann [129], Tryba and Ritzmann [130] and Watson and Ritzmann [131] who have shown for cockroaches (*Blaberus discoidalis*) that the motor neurons control joint velocity. We utilized artificial motor neurons to represent this function (Fig 2, MN). Examples are given in the Supporting information (S1A PDF). The controller of the motor receives these velocity signals as set points for a PID controller. PID control for moving velocity of leg joints is also supported by Levy and Cruse [132], who found that in a standing stick insect the torques of the leg joints may have changed significantly although position of joints is not changed at all, i.e. velocity is set to zero.

The torques developed to reach the desired velocity can approximately be represented by the deviation between the desired joint position and the real joint position which differ due to an elastic element implemented in the motor [81]. In the simulation, the spring constants for the beta joint (see below) vary following the relation 8.5:4.5:8.5 for front leg, middle leg and

hind leg, respectively. Middle legs are equipped with a softer spring to minimize pitch movements (i.e., tipping to front or back) when the middle legs are in stance mode.

The robot simulator, as the physical robot, is run by an update rate of 100 Hz, i.e. an update time of 10 ms. The motor output is adjusted in such a way that the behavior (duration of swing and stance) shows a ratio of about 10:1 compared to that of stick insects.

## The neural Controller

We will explain the neural controller starting with the employed neuron model, which is one major change compared to Walknet. The leg level and the individual joint control will be explained second. Third, coordination between legs will be addressed.

**A) Neuron model.** The neural controller consists of artificial neurons.:

Activation dynamics of neurons are usually approximated by using Hodgkin-Huxley differential equations [133] or related versions [134]. To simplify computation, here we use a reduced version of that equation, similar to that applied by Manoongpong et al. [135], or Schilling et al. [24]. For the discrete time domain this is realized in the python implementation as

$$v_{update}[i] = v[i] + (-v[i] + I_syn[i] + I_app[i])/C_m[i]$$

where [i] denotes the index of the neuron, v the voltage at time t-1, *v_update* the updated voltage at time t, *I_syn* the input from synaptic connections. *I_app* external (e.g., sensory) input. This means the dynamics show low pass filter properties with a time (membrane) constant C_m[i] of 4.5 ms, if not stated otherwise (for details on the implementation see code base). Update time is 1 ms (= 1000 Hz). The simplifications include a piece-wise linear characteristic, i.e. linear but with a lower threshold of 0 mV and an upper bound of 50 mV. Synaptic inputs are summed up linearly and, separately for excitatory input and inhibitory input, are clipped at 80 mV. Only selected units are in addition equipped with a phasic property to allow for adaptation to a constant input, i.e. these units show properties of bandpass filters (marked HPF for high-pass filter in Fig 2). To approximate the neural dynamics, we applied a nonlinear high-pass filter with a high time constant for decreasing voltage (e.g., 10.0s), whereas small time constants (0.01 s) are applied to recover from values below 0 mV, i.e. for increasing voltage. Following Dale's law [136], any unit is either inhibitory or excitatory, i.e. a unit is equipped with either excitatory output synapses or inhibitory output synapses.

**B) Leg control.** Each leg is controlled by a network of about 164 units (plus ring net, Fig 11) and has an identical architecture. A leg controller consists of three subnets (Fig 2, bottom three rows, marked as alpha, gamma, beta), one for each joint, and a number of global units to allow for switching on or off walking and for switching between walking forward or backward, plus the units required for sending and receiving information to and from other legs for inter-leg coordination (Fig 2, brown and ocher units). Further, there is a global input to all joint controllers representing the walking velocity of the hexapod (leftmost white unit–upper part of Fig 2, below coordination units shown in brown). This unit receives input between 0 mV and 50 mV. There is sensory feedback from the joints in form of a simplified antagonistic sensor structure illustrated by the black squares containing piecewise linear functions (input: angular degrees, alpha pos., beta pos., gamma pos., output: neuronal excitation in mV). This structure projects on an artificial neuron allowing for activation between 0 and 50 mV.

Correspondingly, we simplify the motor output by considering two antagonistic muscles for each joint, the protractor-retractor group for the alpha joint, the levator-depressor group for the beta joint and the flexor-extensor group for the gamma joint. In the simulation, each muscle pair is represented by one motor as explained above.

Next, we will explain how sensory input elements and motor output elements are connected.

*Control of Beta Joint (bottom part of Fig 2):* The beta controller represents the most simple case: Each of both antagonistic branches (Fig 2, levator above, depressor below) is designed as a negative proportional position feedback (NPPF) controller: each branch receives its sensory input (joint position value, between 0 mV and 50 mV) that is subtracted from a given set point (rightmost red units for each branch). 'Proportional position controller' means that the negative feedback controller behaves like an elastic spring. The value of the set point depends on the state of the leg (swing or stance; these are channeled from light grey units in upper part of Fig 2) and is given by a small network called height net (beta controller, blue and red units, for details see below). The error signals (output of red units marked "error") are then given to the premotor units (PMN, left, units connected via inhibitory neurons) which in turn project to the motor units (leftmost red units, MN). In all joints, the motor output represents a velocity signal which is transformed, via the motor, to a torque signal. To minimize co-contraction, the premotor units of both branches are connected via recurrent lateral inhibition (for details see below).

Behavioral data [137] suggested the interpretation that, during stance movement, body height, i.e. the distance between body and ground, is controlled by NPPF controllers which operate independently in each leg. In the simulation, we apply a simple network for control of body height. Basically, the beta controller, due to its property to produce proportional error deviations, could already serve as a primitive height controller. However, if the gamma joint adopts large angle values differing too much from a vertical position of the tibia, height control based on the beta controller alone would produce inappropriate set point values. Therefore, signals from gamma angle are also exploited in height net to improve height control. This is done following Fink et al. [138] via presynaptic inhibition to implement a multiplicative dependence (called devisive normalization) via the synaptic input to the central unit of the three rightmost lower red units (Fig 2). S1 Fig in the Supporting information illustrates how the set point value of the beta controller depends on gamma angle in a range of 60 to -57 degrees. The range used during normal walking amounts between -9 and -23 degrees.

In the beta controller, when swing mode is started, a unit with bandpass properties (uppermost large green unit, marked HPF) is activated to "disturb" the position controller in such a way that the levator branch is briefly activated leading to a levation of the leg [78]. As the activation of this unit ceases, the negative position feedback controller pulls the beta joint back to its set position. This "disturbance" signal is applied to agree with the observation that the maximum amplitude of swing movement is constant independent of starting level of the leg [78]. The signal from this unit may also be given to the extensor branch of the gamma joint controller, which helps to lift the tip of the leg higher up during swing. For very slow velocities this "disturbance" is smaller, leading to a short step (see below).

The controller for the beta joint can be used for both forward walking and backward walking.

*Control of Alpha and Gamma Joint*: Alpha controller and gamma controller have to handle forward and backward walking differently. The structure of both alpha controller and gamma controller requires two symmetric branches: Control distinguishes between different behaviors, swing and stance. The mapping from swing and stance to activation of the antagonistic motor neurons is different in forward and backward walking. Similar to Ayers and Davis [139], the activation of the two motor neurons is actually reversed in the case of backward walking with respect to forward walking. Therefore, it is necessary that the input signal for swing as well as that for stance can be given to both motor output units of each joint.

The controllers for swing movements are based in principle on the same structure as the beta controller, i.e. show a NPPF controller. An error signal is calculated from the currently sensed joint position (given by the black squares addressed above) and a set point provided as a target position for the swing movement (set points may vary during negotiation of curves, see Supporting information). In the alpha controller and the gamma controller the error signals (two right large green units in the alpha branch and in the gamma branch) are given to the premotor units via an additional parallel feedforward connection (corresponding leftmost small green units). These connections guarantee a minimum constant output value until the error signal is about zero (i.e., until the joint angle has reached its set point). Note that the set point is beyond the position where end of swing is normally reached as swing is terminated when ground contact is sensed.

Stance movement depends on the global input signal for the velocity represented by a global velocity unit (Fig 2, leftmost white unit). This unit can adopt values between 0 mV and 50 mV (in the simulations we use values from 8 mV to 50 mV). For each leg, this global value is transmitted by a special network ("ring net", Fig 11) that determines the velocity parameter entering the alpha branches and the gamma branches, via the blue units, at its rightmost red units, respectively, but only during stance (see [140]). These signals depend on the alpha position of the leg under view and the angle characterizing the currently desired walking direction theta (equals zero degrees for walking straight forward). This ring network will be explained below, after briefly explaining how switching between the two behaviors takes place.

*End of swing, end of stance*: Before we turn towards coordination of legs, we want to briefly summarize and explain how the single leg controller controls behavior over a single cycle. In particular, the focus shall be on the transitions between the two different behaviors, stance and swing.

Swing movement is finished at the latest as soon as the swinging leg has reached a given anterior position. If, however, the "torque" signal from the beta joint is higher than a given threshold, the leg is considered under load, and this information is used to stop swing even if the leg has not yet reached the prescribed anterior position of the alpha joint. In this case, the rightmost unit of the group of four light grey units (representing state of stance) is activated by the torque signal and protractor motor neuron is inhibited. This mechanism takes however place only if the leg has reached a given position during swing movement as observed in stick insects [78].

Correspondingly, end of stance is determined either by reaching a given leg position in extreme cases or, normally, by earlier signals from any interleg coordination influences. A direct influence of load to suppress swing is not implemented. Instead, stance motivation (via the rightmost turquoise unit) is decreasing versus the end of stance which allows stronger coordination influences to finish stance earlier [141,142]. As a specific property to simplify start of swing, the protractor output is inhibited as long as the leg is already in swing state but still under load. In this way, the leg can be lifted in vertical direction and slipping over ground elicited by possible extension of the gamma joint is minimized.

To summarize: The alpha controller and the gamma controller represent a velocity controller during stance and a proportional position controller during swing using the PID velocity controller as a lower level cascade. In the beta joint, proportional position control is applied during swing and during stance.

*Adaptation of Leg Movement depending on Walking Direction–Ring Net*: To allow for an approximately straight trajectory of the leg tip, the contribution of the three leg joints should be controlled accordingly (in the current approach contribution of angle beta is neglected). In addition, curve walking requires a change of the stance trajectory of the leg (Fig 6). To this end, each leg is equipped with a "ring" network (Fig 11) that controls the alpha joint and the

gamma joint during stance to provide a leg trajectory that contributes to walking in a direction given by angle theta (e.g., theta = 0 for all legs when walking straight forward). This network consists of five layers each containing 12 units (shown from right to left in Fig 11A, Fig 11B gives an overview showing the complete ring structure schematically). Functionally, the units of layers 1–3 (Fig 11B, green ring) act as binary elements. The first layer (Fig 11A, rightmost column, delta) is activated by sensory input representing the angle delta (delta = alpha*–theta, Fig 2, box "spatial coding") with alpha* = (alpha + 50)/3.). Angle alpha is provided by the protractor branch, i.e. the sensor cell representing the alpha angle that increases activation from rear to front. The desired walking direction for each leg is provided by the box "local(theta)" (Fig 2, upper part, below white units). Function local(theta) might be different for different legs depending on the curve radius, i.e. global walking direction. The transformation of rate coded delta to spatial coded delta is not implemented explicitly (Fig 2, box "spatial coding"). Rather, parameters are given for a specific example of curve walking only. Fig 2 illustrates the dependence graphically.

Angle delta is represented by using so-called spatial coding. As 12 units are given for an angular range of 180 degrees, one unit represents an angular range of 15 degrees (Fig 11B). This means that for alpha = 30 degrees and desired direction theta = 0, i.e. straight forward, for example, in the first layer all units from C1 to C6 are excited (see Fig 11A, green dots).

Layer 4 and layer 5 (Fig 11A) control velocity of alpha joint and gamma joint, respectively. The function of layers 2 and 3 is to produce activations in layers 4 and 5 in such a way that only one unit is activated to represent the position of angle delta: First, in layer 2 only one unit is activated, the one that specifies the current delta angle. In the example depicted in Fig 11A this would refer to unit C6 (green dots). Layer 3 represents an inverted version of layer 2 which is required as we need active inhibition of all units in layers 4 and 5 by the units not activated in layer 2 except the one being activated in layer 2. All units of the third layer are then projected to a forth and a fifth layer in parallel. These units receive input from the unit representing the desired walking velocity (Fig 11A, vel; Fig 2, box "local(theta)) via the fixed weights "0.2* sin(alpha)"for layer 4 and "0.2*cos(alpha)"for layer 5. All 12 units of layer 4 (Fig 11B, red) project to output unit #O2 (currently, unit O1 is not yet connected to protractor, as the latter is required for negotiating tight curves only). Units C81 to C86 of layer 5 (Fig 11B, yellow) project to unit O4, units C87 to C92 (Fig 10B, blue) project to unit O3. Units O1 –O4 represent the contribution of the alpha joint (O2) and the gamma joint (O3, O4), respectively, to produce an appropriate leg movement. Units O1 and O2 in Fig 11A correspond to the rightmost dark blue units addressing the alpha branch (Fig 2, upper) and units O3 and O4 to the corresponding units addressing the gamma branch (Fig 2, lower). As depicted in Fig 2, the four left dark blue units of the alpha branch (upper) and the gamma branch (lower) respectively receive–via the small dark blue units–this information and are responsible to direct the signals to the appropriate motor units depending on (i) position of the leg, (ii) the current state, swing or stance, and (iii) forward (including curve walking) or backward.

Taken together, the ring net as used here represents an approximation of a vector with the motor output components of alpha joint and gamma joint pointing to the desired walking direction theta (e.g., straight forward, theta = 0). It is an approximation only because (i) the angle values used have a limited resolution (accuracy 15 degrees) and (ii) the contribution of the beta joint is neglected. Nonetheless, the behavioral results obtained in the simulation show that such an approach is sufficiently able to describe the behavior of the insect (the relation of the ring network to a neural structure found in the central complex is addressed in the Discussion). The ring net can be seen as a simplified version of the body model introduced by Schilling et al. [143], when walking direction is the same for all legs, i.e. for walking straight forward, backward or for holonomic sideways walking. If the function of the body model to

allow for negotiating curves should be realized, too, ring net has to be complemented by an additional system, local(theta), that is able to provide the direction of individual legs.

As a general property, not explicitly required for straight walking, we added a limitation of stance movement not only with respect to the alpha angle, but also to the gamma angle (upper limit = 35 mV, Supporting information, S1 Fig). This is required for negotiating curves as, due to the oblique trajectory, the inner front leg would not reach the alpha thresholds before the leg had moved below the body (see Fig 6). The existence of such limits follows from data given by Dürr and Ebeling [57] and also by Gruhn et al. [59].

For a proof of concept we then tested a specific case using a turning angle theta of 75 degrees. How are directions of front leg trajectories influenced? For control of swing movement, the beta controller is used as during forward walking, but the set points of the front legs are now specified correspondingly. The set points for the AEP of alpha joint and gamma joint appear to depend on angle theta. As detailed experimental result are only available from Dürr and Ebeling [57], we chose plausible set point values that approximated their data: for the inner front leg alpha = 20 mV, gamma = 20 mV, for the outer front leg alpha = 50 mV, gamma = 45 mV. During stance, the contribution of alpha joints and gamma joints are determined by the ring network (see above) as in forward walking, i.e. by angles delta = alpha$^*$– theta. In the case of curve walking, theta deviates from zero. Following Rosano and Webb [60], for the inner front leg theta is set to point to the desired walking direction, e.g., the direction of a goal to be approached. For the specific case shown here, we set theta = 5 (corresponding to an angular range of 75 (± 7.5) degrees). As a first approximation to data of Dürr and Ebeling [57] and Rosano and Webb [60], we reduced theta for the outer front leg by a factor of 0.25.

How are the specific local velocities determined? To approach the specific data given by Dürr and Ebeling [57] we adapted global velocity input (in the example 45 mV) by factor of 1 for all three outer legs and factors of 0.75, 0.35 and 0.1 for the inner front leg, middle leg and hind leg, respectively.

For simulation, all these parameters (position of AEP, theta, local stance velocity) have been stored in "local(theta)" (Fig 2) and are given to the swing set points of both front legs for swing and the local velocities as well as the local delta values to the ring net of each of the six legs, correspondingly, for stance.

Two types of short steps have been observed [32,89]. One type can be observed if, at the end of swing, the tarsus does not find efficient adhesion to the substrate. Therefore, the leg may react by lifting the leg up and down similar to that of a searching movement. As this type of short step may be caused by the same structure as used for searching, we did not simulate this behavior. Instead, we implemented another type of short step. During very slow walking, while trying to search a way to cross a gap, stick insects (*Aretaon asperrimus*) may perform swing movements showing a very small amplitude [32]. This behavior could be simulated by neuro-Walknet if, for very slow walking velocities, the output of the uppermost large green unit (Fig 2, two uppermost green units marked "short step") is decreased. Thereby the levator would be lifted to a small degree only resulting in a short swing movement. This type of short step emerges in the inner hind leg when negotiation of tight turns is simulated.

**C) Interleg coordination.** Behavioral observations in forward walking stick insects have led to five "rules" that may underlie interleg coordination [15]. In these experiments various disturbances, essentially prolongation or shortening of swing duration or stance duration, have been applied to animals walking either unrestrained, or supported by a holder on a treadwheel or on slippery surface. Here we deal with rules 1–3 first (all units marked brown, Fig 2; these have been realized in Walknet as well, Schilling et al. 2013a) and later rule 5 (all units marked "ocher", Fig 2; this has been realized in an earlier version of Walknet [86]). We will not test rule 4, which describes the faculty of one leg to use the position of its anterior

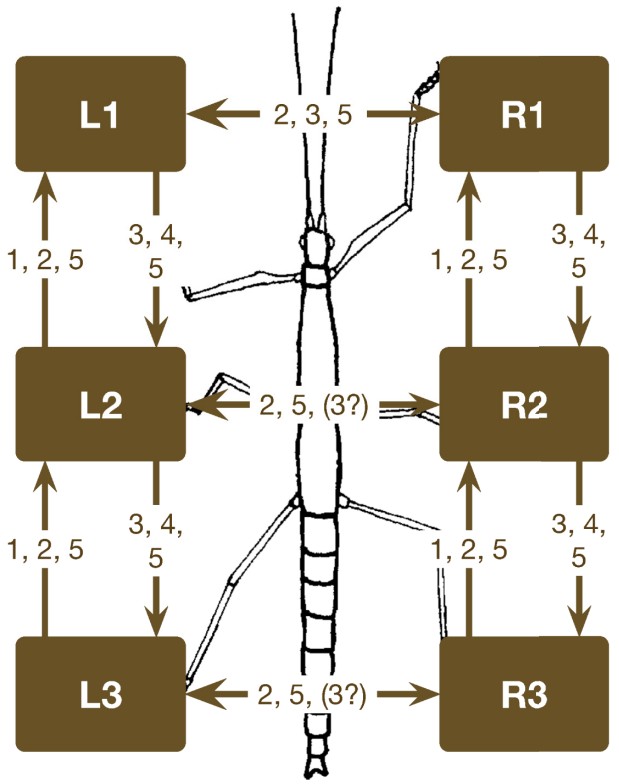

1. Return stroke inhibits start of return stroke

2. Start of power stroke excites start of return stroke

3. Caudal positions excite start of return stroke

4. Position influences position at end of return stroke ("targeting")

5a. Increased resistance increases force ("coactivation")

**Fig 12. Single leg controllers (shown in brown) and their connection via coordination rules (from [15]).** L1, L2, L3 left front, middle, and hind leg, respectively. R1, R2, and R3 stand for the corresponding right legs. The question mark indicates that there are ambiguous data concerning influence 3.

neighboring leg as a "target", or set point, for the goal of its swing trajectory. One reason for that is that the size of front legs and hind legs in the robot do not fit to those of the stick insect, making simulation of targeting behavior difficult. Another reason is that it is not known to what extent targeting may be activated or not (see Supporting information). We also do not address interesting results showing that coordination between legs is also driven by mechanical coupling between legs [33] which may support–or replace–rule 1 as a neural mechanism, at least when walking on horizontal substrate.

All rules connect only directly neighboring legs and depend on state (swing, stance) of the sender leg as well as on the state of the receiver leg (see Fig 2, Fig 12). In Fig 2 the connections to and from other legs are depicted by bold dark arrows.

*Rule 1* is assumed to act only between ipsilateral neighboring legs in anterior direction (rear to front). It inhibits start of swing in the anterior leg and is split into two sections (1a, 1b). Influence 1a is active as long as the sender leg is in swing state (brown units, output 1a). Influence 1b represents a temporal delay driven by an integrator (characterized by a recurrent self-activation), the output of which is however inhibited when rule 2i is activated, too (see below).

*Rule 2i* (i stands for ipsilateral) depends on velocity via the rightmost of the three lower brown units inhibiting the output of the integrator. As soon as inhibition of this unit is finished, two serially connected units (marked by "HPF", together forming a second order high pass filter) activate an output signal that represents rule 2i. This effect may excite swing in the receiver leg, so that inhibition of swing via 1b is directly followed by excitation of swing. In

other words, rules 1 and 2i together may trigger swing in the anterior neighbor in such a way that, after a short delay following the end of swing movement of the sender, swing of the receiver leg is started. For high velocities the delay is short, but increases for slower velocities (see [34]).

*Rule 3i*: Complementary to rule 2i there is rule 3i, as it elicits swing, too. However, rule 3i operates ipsilaterally from a sender leg to its posterior neighbor (i.e., from front leg to middle leg, and from middle leg to hind leg). This influence depends on alpha position of the sender leg and switches on (thr_on) or off (thr_off) the sender signal which depends on leg position as well as a threshold function depending linearly on velocity. As both influences, 2i and 3i, superimpose in the case of the middle leg, parameters should ideally be chosen in such a way that 2i being sent from the hind leg starts together with 3i being sent from the front leg so that both influences support each other. However, longer duration of 3i activation may act to move not yet optimally coupled legs nearer to optimal values.

*Contralateral–rule 2c and 3c*: Further, there are two influences 2c and 3c (c for contralateral) that act between contralateral neighbors and act symmetrically in both directions. Influence 2c is dependent on the position of the sender leg via the unit marked "HPF". Rule 3c depends on position of the sender leg, too, but in this case there is no high pass filter. This means that the effect of 2c decreases with decreasing velocity, in contrast to influence 3c.

*No Rule 1 contralateral*: The effect of rule 1 has first been postulated by Hughes [48]. First indirect experimental evidence pointing to this effect has been found for Neoconocephalus between hind leg and middle leg by Graham [144] and indirectly for contralateral legs in *Carausius* [145]. Direct experimental evidence has however only been revealed among ipsilateral legs (hind leg to middle leg, middle leg to front leg), but not for contralateral legs [146]. Therefore, rule 1 influences for contralateral legs have not been implemented.

*Backward Walking*: For backward walking, rules 1 and 2i are used in the same way as during forward walking. Rule 2c for backward walking corresponds to rule 2c for forward walking, with a however different position threshold. Rule 3i and 3c are not activated during backward walking as we searched for a minimal number of parameters.

*Rule 5*: In addition to rules 1–3, rule 5 (5i, 5c) is implemented, too (here we refer only to rule 5a, not to rule 5b, Fig 12). Rule 5 differs fundamentally from the other rules as it depends on the output of the retractor premotor unit of the sender leg (shown as the upper leftmost unit marked in ocher, output 5) and that it excites the retractor premotor unit of the receiver legs while at the same time inhibiting the antagonistic protractor premotor units (for other joints, see below). In other words, these influences support cooperative (in-phase) motor output of legs being coupled. In Fig 2 input via rule 5i is depicted as the upper rightmost unit in ocher. The left neighboring unit receives input 5c. Rule 5 acts between all neighboring legs except both hind legs [92]. The influences are only effective if the retractor of the sender is strongly activated. In other words, during undisturbed walking rule 5 influences are not activated. For the hind leg, we further introduce rule 5ch (leftmost unit of rule 5 input), which shows the same kind of connectivity, but inhibits the retractor and excites the protractor premotor units of the receiving leg, i.e. supports anti-phase coupling, an assumption strongly inspired by results of [69].

Several studies concerning the coupling of CPGs [54,67–69,72] reported comparable results for recordings from retractor motor neurons (i.e., alpha joint) and from depressor neurons (i.e., beta joint). As a consequence, rule 5 has been introduced to both joint controllers. As Büschges et al. [67] showed that oscillations of alpha joint and beta joints are not coupled with each other, both joints are connected independently via rule 5.

A number of behavioral and neurophysiological studies support the assumption that coordination between ipsilateral legs is stronger than between contralateral legs (for walking, e.g.,

[76], for deafferented preparations e.g., [69]). To agree with this property, rule 5 connections between ipsilateral legs are coupled by a factor of 0.2, whereas connections between contralateral legs are connected via a factor of 0.1.

## Specific experimental settings

The simulations have been applied to simulate a wide variety of different locomotion experiments. In these cases, different velocities were used and, for moderately fast velocity, curve walking was tested. Furthermore, we performed simulations that parallel specific experimental settings: on the one hand, experimental setups in which animals were put on a treadmill where in some cases only a couple of legs were walking. On the other hand, experimental interventions in which the animal was deafferented and as a consequence there was no sensory input from the animal nor movement produced. In the following, we will describe how we simulated these experiments.

**Setup of treadmill experiments.** In some experiments [72,73,94] animals were fixed on a holder and walked on a treadmill, whereas other legs were deafferented or were intact but standing on a fixed substrate. We assume that rule 5 influences of legs walking on a treadmill are stronger than leg controllers of a ganglion ´treated with pilocarpine´.

Simulation of motor output required for a single leg moving a treadmill is not straight forward because the forces to be applied by the animals may vary depending on the inertia and friction of the specific treadmill applied. Even the extremely light-weight treadwheel developed by Graham [147] still had a two-fold larger inertia compared to a free walking stick insect in the horizontal direction and an eight-fold inertia in vertical direction. Applying an indirect measure, Cruse [148] could show that front legs show stronger forces on a treadwheel, but no data are available concerning contribution of alpha or gamma joint. Therefore, in order to reproduce–on a qualitative level–results observed by the experiments of Borgmann et al. [72] and by Cruse and Saxler [94], we use the following approximation. Legs walking on a treadmill are assumed to produce higher rule 5 signals at the beginning of stance, due to higher friction to be compensated (see [92]). To this end, during the first half of leg position (alpha joint position sensor > 25mV) the factor of rule 5 is increased from 1 to 15, but, to compensate friction effects, does not increase velocity output. This is necessary because friction cannot be simulated in the robot Hector. In the figures, MN output activation is shown if above a given threshold (see legends of Figs 8–10, Supporting information S4 Fig).

**CPG Experiments with a deafferented system.** Although the general architecture of neuroWalknet does not rely on CPGs to control normal leg movement, the network contains elements that under specific conditions may operate as oscillators.

As mentioned above in this section, the antagonistic branches of each joint controller are connected via lateral inhibition to avoid co-contraction. A network consisting of two units connected via lateral inhibition could be made to oscillate, if (a) we deal with backward (recurrent) inhibition rather than forward inhibition, (b) if inhibitory connections are strong enough, (c) if recurrent inhibitory units show adaptive (i.e., high pass filter like) properties, and (d) most important, if both units receive parallel excitation. In our case the latter condition is normally not given during normal walking, as we deal with an antagonistic architecture, where in general only one of both channels is activated. However, if sensors are either deafferented or, in addition, treated with pilocarpine, both channels are activated and therefore oscillations can be elicited [67]. Details of how this neuromodular effect operates are not known. We decided to apply this effect as hypothetical excitatory input to those neurons marked "pilo" in Fig 2. This may reflect stimulation of campaniform sensilla, but any other mechanosensor may be suited to elicit this function, too. As there are no reports concerning mutual inhibition

to exist between motor neurons, these connections are implemented on the premotor level. The fictive sensors are implemented for each joint but used only to activate both premotor units to allow for simulation of treatment with pilocarpine, i.e. respond to strong activations only. To simulate experiments with deafferented (e.g., autotomized) and/or pilocarpine treated animals, we fix the motor output (joint angle velocity) to zero in all joints, and set all inputs to the PMN units, marked by "pilo" in Fig 2, to high values ($>\ =$ 25 mV).

## Supporting information

**S1 Fig. Body height depending on angles beta and gamma.** Body height depends on angles beta and gamma (the contribution of angle alpha is neglected). The network controlling body height during stance is depicted in Fig 2 (three blue units, lower right) and is actively controlling the set point of the beta angle (see Methods, Control of beta joint). Here it is shown how, during stance, the set point for the beta joint controller depends on the gamma angle (see inset visualizing how both define body height). This function is approximated by presynaptic inhibition (Fig 2, input to the central unit of the three blue units, lower right, as depicted by the blue dots). Abscissa: gamma angle (degrees) and flexion (mV), which is used as input for height control. Ordinate: set point for beta controller during stance (mV), as depicted in the inset. The red bar marks the range of gamma angles used in the simulations during normal walking. (EPS)

**S2 Fig. Additional footfall patterns for additional walking velocities.** More footfall pattern for forward walking, velocity neuron set to 45 mV (A), 37.5 mV (B), 35 mV (C), 32.5 mV (D), 12.5 mV, (E). (EPS)

**S3 Fig. Additional footfall patterns–emerging patterns.** Emergence of a stable footfall pattern is shown in (F), (G) as in Fig 3, but now transients are shown, when starting with the usual leg configuration. Velocity neuron set to 10 mV (F), for 240 s, and 8 mV (G), for 240 s, for stable pattern see Fig 3H. Ordinate: legs as in Fig 3, abscissa: time (s). (EPS)

**S4 Fig. Simulation of the Borgmann et al. (2009) experiment.** Simulation of Borgmann et al. (2009) as in Fig 9, but using different velocities for the walking leg (FL). All other legs are deafferented and treated with pilocarpine. Velocity neuron set to 40 mV (A), 20 mV (B). Ordinate: legs as in Fig 3, abscissa: time (s). (EPS)

**S1 Document. Sections of steps for different legs.** (PDF)

**S2 Document. Results and adaptation of model for running.** (PDF)

**S3 Document. Differences between species concerning interleg coordination and chosen coordination influences.** (PDF)

**S1 Video. Forward walking, velocity neuron set to 15 mV.** Simulation run of the robot, footfall patterns see Fig 3. (MP4)

**S2 Video. Forward walking, velocity neuron set to 20 mV.** Simulation run of the robot, footfall patterns see Fig 3.
(MP4)

**S3 Video. Forward walking, velocity neuron set to 25 mV.** Simulation run of the robot, footfall patterns see Fig 3.
(MP4)

**S4 Video. Forward walking, velocity neuron set to 30 mV.** Simulation run of the robot, footfall patterns see Fig 3.
(MP4)

**S5 Video. Forward walking, velocity neuron set to 35 mV.** Simulation run of the robot, footfall patterns see Fig 3.
(MP4)

**S6 Video. Forward walking, velocity neuron set to 40 mV.** Simulation run of the robot, footfall patterns see Fig 3.
(MP4)

**S7 Video. Forward walking, velocity neuron set to 45 mV.** Simulation run of the robot, footfall patterns see Fig 3.
(MP4)

**S8 Video. Forward walking, velocity neuron set to 50 mV.** Simulation run of the robot, footfall patterns see Fig 3.
(MP4)

**S9 Video. Simulation driven at high velocity including CPG activation see S2 PDF).**
(MP4)

**S10 Video. Curve walking.** Simulation run of the robot, details on foot trajectory and footfall pattern see Fig 6.
(MP4)

**S11 Video. Backward walking, velocity neuron set to 20 mV.** Simulation run of the robot, footfall patterns see Fig 7.
(MP4)

**S12 Video. Backward walking, velocity neuron set to 30 mV.** Simulation run of the robot, footfall patterns see Fig 7.
(MP4)

**S13 Video. Backward walking, velocity neuron set to 40 mV.** Simulation run of the robot, footfall patterns see Fig 7.
(MP4)

**S14 Video. Backward walking, velocity neuron set to 50 mV.** Simulation run of the robot, footfall patterns see Fig 7.
(MP4)

## Acknowledgments

The authors thank Thierry Hoinville, Josef Schmitz, Axel Schneider, and Volker Dürr for discussions and helpful comments on the manuscript.

## Author Contributions

**Conceptualization:** Malte Schilling, Holk Cruse.

**Data curation:** Malte Schilling.

**Formal analysis:** Holk Cruse.

**Funding acquisition:** Holk Cruse.

**Investigation:** Malte Schilling.

**Methodology:** Malte Schilling, Holk Cruse.

**Software:** Malte Schilling.

**Writing – original draft:** Malte Schilling, Holk Cruse.

**Writing – review & editing:** Malte Schilling, Holk Cruse.

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
