## [Decision Letter · Decision Letter 0]

23 Sep 2019

Dear Dr Schilling,

Thank you very much for submitting your manuscript 'Decentralized control of insect walking - a simple neural network explains a wide range of behavioral and neurophysiological results' for review by PLOS Computational Biology. Your manuscript has been fully evaluated by the PLOS Computational Biology editorial team and in this case also by independent peer reviewers. The reviewers appreciated the attention to an important problem, but raised some substantial concerns about the manuscript as it currently stands. While your manuscript cannot be accepted in its present form, we are willing to consider a revised version in which the issues raised by the reviewers have been adequately addressed. We cannot, of course, promise publication at that time.

Sincerely,

Joseph Ayers, PhD

Associate Editor

PLOS Computational Biology

Samuel Gershman

Deputy Editor

PLOS Computational Biology

[LINK]

Reviewer's Responses to Questions

**Comments to the Authors:**

Reviewer #1: The study of Schilling and Cruse addresses a timely topic in the field of motor control, i.e. how control architectures can contribute to the generation of a highly adaptable walking behavior with respect to the generation of a broad range of stepping velocities. In the course of the study, the authors present a new control architecture, neuroWalknet, extending the well elaborated simulation Walknet. Using this simulator the authors show potentially very interesting data on the generation of interleg coordination patterns changing with walking speed of the simulator and interesting evidence for their controller architecture to generate motor activities similar to those found in deafferented and pharmacologically activated thoracic ganglia of insects. Such study might be interesting for a broad audience of scientists in the field of robotics and motor control.

However, in its present form the manuscript shows shortcomings that need to be dealt with by the authors in order to allow a reader to follow and assess the results. I have therefore only listed my most major criticisms.

1. The manuscript is on the one hand far too extensive, even for an informed and highly interested reader. On the other hand, it still lacks important information for backing the conclusions drawn by the authors – major points 2 - 4 potentially directly arise from this current drawback. The reason for this situation appears to be that the authors not only present data on their three research questions in focus based on new simulation studies, but also present a completely new simulation of the stick insect walking system. Furthermore, neither the introduction, nor the discussion is well focused. This is in particular obvious from the discussion, in which also issues are discussed, which have nothing directly to do with the present study (but perhaps with the new simulation), for example, the generation of searching movements (line 665 and following) and the existence of the ring networks in the simulation (line 969 and following). I strongly suggest that the authors consider dividing this manuscript at least in two, one presenting the new simulation together with the necessary detailed presentation of its composition and its testing and a second dealing with the three present questions in focus.

2. One concern is directed towards the concept of the simulation (line 271 and following): given today’s knowledge about the relevance of muscle properties for the generation of movements and the existing wealth of literature it appears puzzling that the authors generate a dynamic simulation that essentially leaves this most important output level of movement generation unrepresented. Detailed information on muscle properties is available for the simulated organism, i.e. the stick insect, making it with respect to muscle properties one of the most thoroughly studied insect species in that respect. This shortcoming is even more relevant as the authors argue that for their research concept closing the loop through the environment is crucial and as they aim with their simulation for a holistic description based “on a minimum of a priori assumptions” allowing for a more complete testing of conclusions drawn from experiments (line 286 - 298).

3. Another concern addresses the role of network components of the simulation (Introduction and Figure2): In the introduction the authors try hard to relativize the functional significance of the ubiquitous finding that central neural circuits generating locomotor activity in many (if not all) studied rhythmic locomotor forms, i.e. swimming, flying, walking, and crawling, contain elements or networks that can generate rhythmic activity without any phasic input, so-called CPGs. One part of their study aims for providing evidence, how their decentralized neural control structure can explain neural oscillations without an explicit CPG network organization involved (at least for low cycle periods). When inspecting the topology of the simulation presented in Figure 2, it becomes immediately clear that each joint control network contains the architecture for a CPG in the circuit connecting the PMNs via mutual inhibition, a circuit scheme ubiquitously found in CPGs and assuring the generation/maintenance of phases of activity (such minimal network will assure a two-phasic activity). The network topology used here represents the ideal form of a so-called half-center CPG. Please comment and provide evidence that the simulation also works, when removing these CPGs.

4. Except for one “experiment” (Figure 5) the authors only provide exemplary figures of results of the simulation and practically no quantitative data. This unfortunately renders their conclusions presently as claims only. This applies for all sections, i.e. “Forward walking – emerging gaits”, “Backward walking”, “Analysis of neural activity – the possible function of Intrinsic Oscillation”.

Here are some selected examples out of many more instances of the study for relevant issues to be addressed with quantitative data.

Line 368 - 431: At the moment, for example, it is not clear, how the authors can come to the conclusion that their simulation is generating a continuum of gaits with increasing voltage. Also, the fact needs explanation that the maximal stepping frequency for a single leg apparently does not exceed 0.2 Hz (Figure 3A), which is even by far slower than a stick insect walks.

Line 432 - 438: How does backward walking result from activating rule 1 and 2i, what other solutions were tested and what is the experimental evidence to select these two rules?

Line 472 - 506: What values were tested for the load sensors and what was the resulting activity of the premotor neurons? How can be excluded that the results are a consequence of the built-in CPGs (see above)?

Reviewer #2: This excellent paper presents a sensory based controller for insect walking that does not contain a central pattern generator. This controller utilizes information from sense organs to produce walking patterns and is capable of generating a wide variety of gaits, similar to those seen in insects. Furthermore, the controller effectively resists perturbations, is capable of rapid stops, changes in velocity and can walk both forward and backward.

The manuscript is remarkable for the clarity of the writing and logical expositions. This paper will be of considerable value to the readers of PLOS, including neurobiologists, engineers and roboticists. My only comments are the addition of some references and minor changes that I believe would benefit the reader.

Additional Reference

1- Walking/running requires pattern generator - The authors have very clearly and convincingly stated their arguments. However, it is important to remember that CPGs are defined experimentally as elements that produce rhythmic activities in the experimentally-induced absence of sensory feedback. What is being studied only shows the capabilities of the system that remain after injury. It is likely that the identification and characterization of those elements depend upon the experimental procedures producing the isolation. This was very clearly stated by Kier Pearson in a review article years ago and it would be beneficial to the reader to cite this paper (below).

Pearson K.G. (1987) Central pattern generation: a concept under scrutiny. In: McLennan H., Ledsome J.R., McIntosh C.H.S., Jones D.R. (eds) Advances in Physiological Research. Springer, Boston, MA

Minor Corrections

Abstract, beginning line 8 - Redundant use of the words pattern and include; also the term faculty is somewhat unclear. I suggest this would be better as: This includes the ability to produce footfall patterns such as velocity dependent “tripod”, “tetrapod”, and “pentapod” patterns, as well as various stable intermediate gaits as observed in stick insects and in Drosophila.

line 40 - Counterproductive - I believe the authors mean 'counterproductive in robotic controllers'. Evolution does not necessarily exclude neuronal elements based on their efficiency or apparent productivity.

line 76 - Better as 'In contrast to' rather than 'Different' to.

line 145 - It is also possible that, during rapid walking, sensory inputs in one phase could influence the output in the next phase. This question has not been investigated in insects but the authors might consider and mention this possibility. It might also benefit the reader to cite the study of Holtje and Hustert (below) showing that the conduction delays from sensory receptors indicating leg contact/force located on proximal leg segments are very short and could function at higher stepping frequencies.

Höltje M, Hustert R. Rapid mechano-sensory pathways code leg impact and

elicit very rapid reflexes in insects. J Exp Biol 206: 2713–2724, 2003

Reviewer #3: This manuscript discusses a long-standing issue in the control of legged locomotion: are there central neuronal circuits that produce and drive rhythmic leg movements or are there simpler solutions? In their studies, Schilling and Kruse provide a network of neurons that is based on behavioral data from insect locomotion and controls a simulated six-legged robot at varies walking speeds. This network does not feature any rhythmic central components, but instead relies mostly on sensory input and coordinating influences between the different legs. By testing this network in a variety of conditions that have been explored both behaviorally and neurophysiologically they demonstrate that solutions to all challenges emerge from their network architecture, and that these solutions correspond well to previously published locomotion findings. In addition, emerging from the network responses, similarities to known neuronal structures in insects and other legged animals become obvious.

This manuscript thus provides a compelling alternative to current views on how rhythmic leg movements are produced. It is very well written, albeit somewhat long and redundant. I have no major comments on the scientific conclusions, but several that pertain to the order and logic of the arguments presented. In addition, I have quite a few suggestions that may help to improve the reader's understanding of the text (since this is a very long manuscript).

Redundancies and logic/order:

1. Lines 134 and 157: the continuum of stepping patterns / the continuous transitions into other patters. I think this needs to be elaborated more since this is a concept that is later tested explicitly in the model, but it is not introduced well. How are these transitions continuous? Do patterns flip forth and back between different gaits, for example?

2. Lines 176 - 179: switch the order and number them (1 = how to couple, 2 = how to control individual legs). This then fits the order of the next paragraphs, which also should be numbered.

3. Lines 225 - 233: This paragraph seems partly unclear. Is there a way of improving the description of the difference in phasing?

4. Line 250. Delete this reason, it is not necessary and diminishes your arguments. Similarly, the mentioning of the reasons not to address the Manoonpong study sidetracks and distracts. Also, this helps shortening the introduction.

5. One thing missing though is the coordinating influences that are later mentioned in the methods section. Since the 5 influences are a substantial and necessary part to understanding the approach, they should be defined in the introduction, and not be hidden in the methods. This is particularly important for readers outside of the stick insect world that are not familiar with walknet or the original Cruse publications.

6. Fig. 5: There are several dashed lines. Mention which ones are which.

7. Curve walking, line 422: This paragraph needs to be extended. The purpose of the curve walking test is not clear. Most of the pertaining explanations are found in the supplement. However, the purpose of a supplement is not to provide an additional results part, but instead to supplement a fully understandable and complete dataset that is supplied in the results section of the manuscript. The supplement should provide additional information, not the actual main data for the results.

8. Backwards walking: the results are essentially absent for backwards walking (except for pointing at the figure). The results need to be elaborated in the text.

9. Redundancy: Lines 445ff: most of this has been stated before and can be cut. The next page (lines 452 - 475 could be shortened substantially. Is it really necessary to provide details of the previous publications?

10. Lines 544: it is unclear whether this refers to your study or previous experiments. Later in this paragraph: why was the slipping legs chosen as approach?

11. Line 631. No this is not the conclusion from these arguments. Would the conclusion not be that the new controller corresponds well to established behavioral data?

12. Line 640, first part: I do not understand what that means. Next sentence: which levels? Next sentence: can be deleted (shortened). Next para: line 643: delete (repetitive).

13. Lines 653 - 660, then 661 - 664: I suggest to reverse the order in these paragraphs (experimental findings first, then model) to match with the subsequent paragraph and to make logic clear.

14. Somewhere in the discussion I think the old Baessler paper Bässler, U. (1986). On the definition of central pattern generator and its sensory control. Biological Cybernetics, 54(1), 65-69 should be mentioned. It provides a nice discussion about CPG vs sensory control.

15. Lines 689: the purpose of this paragraph is unclear.

16. Line 804: please add what the conclusion is "suggesting that...."

17. Line 857. This paragraph is largely redundant to other parts of the manuscript. It can be shortened substantially while still providing the conclusion.

18. Line 896, conclusions. Again, too much redundancies and repetitions. I understand you want to make sure the main points are understood but at this point of the manuscript it is unnecessary to repeat the arguments for your conclusions. Just providing conclusions is fine.

19. Line 999 ff: "piece-wise linear with a threshold of 0". It is unclear what that means and why this was done given that these are HH models. Similarly, how can a HH model work with no leak? Later: time constants do not rely on leak settings and leak does not have a time constant. This section needs to be improved.

Other comments:

1. Line 52: replace "a" with "some"

2. Line 69: there are quite a few rhythmic biological movements that are driven by well characterized CPGs (with spontaneously active endogenous bursters) that are slow. Those may not be involved in locomotion, but nevertheless drive slow rhythmic movements and should thus be considered in the text here. Invertebrate heartbeat or digestion would be examples.

3. Line 74: Water and air can be quite turbulent and 'unexpected'. It is not clear how that contrasts to walking. Please elaborate in the text.

4. Line 77: did you mean "any joints in a single appendage"?

5. Line 107: "tarsi during climbing"?

6. Line 121: "leads to" or "requires" (or both)?

7. Line 124: replace "and" with "with"

8. Line 147: In the end you are discussing faster speeds as well so this limitation can be deleted from the text.

9. Line 169: replace "any" with "every"

10. Line 184: "which is based on ... ... controllers and coordination influences that mainly regulate..."

11. Line 210: delete "not considered in walknet"

12. Line 247: No, this paragraph lays out questions for the coordination of appendages, not for the functional properties of the CPGs.

13. Line 266: ", the simulation requires a body..".

14. Line 270. Reference Fig 1B here.

15. Line 289: "concern" no "s"

16. Line 348: the name 'neurowalknet' needs to be introduced earlier (NOT in the methods section).

17. Line 369: replace "and with" with "but at"

18. Line 382: add "only" after "reached"

19. Line 402: Explain in the text why it is important to note the average.

20. Line 418: explain in the text which "effort" is meant.

21. Line 434: in line with your other arguments - why is backwards walking now a different state (and not one that a single controller could achieve)?

22. Line 437: Github: May benefit from a better description of the individual movies on the website itself.

23. Line 452: jargon. I believe you meant "deal with THE RESULTS of a study" (not the study itself).

24. Line 472: did you mean "to mimic deafferentation, we cut..."?

25. Line 507: "interesting" or "excellent" studies. While I agree that this is true, I think IF you say they are interesting or excellent, you need to mention WHY this is the case, not just that they are. An alternative would be to just delete these words. For line 507: maybe instead just say that your neurowalknet even handles rather artificial situations that animals had been exposed to previously in experimental conditions.

26. Line 535: upon? Later: "i.e. only some legs were able .."

27. Line 556: why is it important to note that? Similarly: line 583.

28. LIne 615. Provide reference for walknet

29. Line 617. Shorten: Delete "as has been shown"

30. Lines 622 - 625: check the sentences there. Why eventually?

31. Line 634: delete "highly influential"

32. Line 651: did you mean "thereby predicting (or providing) ideas as to how..."?

33. Lines 669: check for redundancies.

34. Line 682: no information is given... What does that mean?

35. Line 685: Replace "in this" with 'consequently, the ...". Delete the "(not simulated) and... " part of the sentence - it does not any important information.

36. Line 721: WTA definition must be here. There is much room in this paragraph to shorten the text.

37. Line 745: CTr??

38. Line 747 ff: shorten. jargon: "deal with"

39. Line 765: "for slow walking CPGs are not necessary, but instead may cause rather nega..."

40. Line 776: no results shown. So what then? Why do you mention it?

41. Line 779: In this paragraph and the next: Check for redundancies.

42. Line 816: "INdependent"

43. Line 821: I very much appreciate this argument.

44. Line 846: Can you predict the outcome of these experiments then?

45. Line 849: does this refer to chaotic systems? If so, mention it and provide reference.

46. Line 877: "later" what did you mean with that?

47. Line 952: which "technical' reasons?

Reviewer #4: The manuscript titled “Decentralized control of insect walking – a simple neural network explains a wide range of behavioral and neurophysiological results” by Schilling and Cruse describes a sensor-driven controller, called neuroWalknet, that is composed of artificial neurons, some of which are meant to simulate the hypothesized functional roles of identified cells in insect nervous systems (mainly Drosophia and cricket) that appear to participate in various forms of hexapod locomotion. The authors make clear more than once that these sensor-based control mechanisms are presumed to play a dominant role mainly in slow walking and exploratory behaviors, while central pattern generators are required for faster and more tightly coordinated forms of locomotion. The manuscript is generally very well written, and the results contribute significantly to the understanding of locomotor control in arthropods. I have only a very few minor comments.

Minor comments:

[Introduction, pg. 3, lines 74-76] Change to: Swimming and flying are operating in fairly homogenous, fluid media. Walking, in contrast, has to deal with a more difficult substrate, providing unpredictable… In contrast to swimming and flying, walking is characterized…”

[Introduction, pg. 4, line 87] “…will begin with a behavioral level of analysis, highlighting the fact that walking comprises…”

[Introduction, pg. 4, lines 92-94] Shouldn’t this section be subtitled “A continuum of locomotor behavior?” Among arthropods, are all forms of locomotion on solid surfaces called “walking”, regardless of varying speed and differing gaits? Is “running” really classified as a very fast form of “walking?” I suggest that lines 93-94 should read something like… “Insect locomotion on solid surfaces spans a whole behavioral continuum (refs) – from slow walking to running, exploratory stepping, gap crossing, climbing, and turning.” I don’t agree with using the word “walking” twice in the first sentence here with two different meanings – walking as a slow stepping gait, vs. walking as a general term covering all terrestrial locomotion.

Reviewer #5: General comments

The authors describe an artificial neural network controlling different walking patterns of a hexapod with 18 DoF. The structure of the neural control system includes local rules coupling different legs through sensory feedback loops; no central oscillatory network setting a common locomotor rhythm and pattern is included in the network. Computer simulations demonstrated a wide range of possible locomotor patterns generated by the robot. The author concluded that such a decentralized control based on the local inter-leg connections via sensory feedback rules without a CPG are sufficient to generate walking for almost the entire range of walking speeds, except very fast ones.

The manuscript nicely fits the scope of the journal and is of interest to a wide range of researchers in the field of locomotor control of biological and robotic systems. The manuscript is well written with detailed rationale for the study, description of the relevant methods, results and discussion. Although the authors’ conclusion that the proposed neural network is capable of producing hexapod various walking patterns appears well justified, their claim that the same general neural network architecture might underlay locomotor control in legged animals is less convincing. The novelty of the proposed idea of decentralized locomotor control seems to be questionable. The literature review in the manuscript seem missing important relevant publications that might challenge authors’ interpretation and conclusions. These and other weaknesses detailed below need to be addressed.

Major comments

1. The claim that the proposed neural network structure can underlay locomotor control of legged animals (i.e., no CPG involvement for most of walking speeds or sensory feedback plays a subordinate role in studies proposing the main contribution of CPG) must be better justified or modified (lines 879-893). Such characterizations of studies supporting the CPG-based architecture and the role of sensory feedback in modulating CPG activity do not seem accurate. It has been well established since the 1920s-1930s by T. Graham Brown (see his unpublished videos of decerebrate walking cats: https://www.youtube.com/watch?v=wPiLLplofYw) and by Grillner with colleagues in 1970s-1980s (e.g. Forssberg et al. Acta Physiol Scand 108: 283-295, 1980) that the decerebrate and spinal cats can initiate and stop locomotion, change gaits from very slow stepping to galloping, or change locomotor rhythms differentially for the left and right hindlimbs based on sensory feedback exclusively. Similar changes in rhythmic activity patterns occur at different intensities of the locomotor region in midbrain of cat fictive locomotion preparations (no motion-dependent feedback). There are detailed CPG-based modes that predict gait changes from very slow walking to fast running based on a simple supraspinal velocity-related signal; the locomotor patterns and gait changes predicted by these models match well experimental recordings in genetically modified and wild type mice (e.g. Danner et al. Elife 6: 2017; Ausborn et al. Elife 8: 2019). In addition, fictive locomotion patterns (no motion-dependent sensory feedback) demonstrate a wide range of swing and stance durations covering a wide range of the corresponding walking speeds from very slow walking to fast running (e.g., Yakovenko et al. J Neurophysiol 94: 1057-1065, 2005).

2. The described neural network does not seem to have postural reflexes that would resist sudden perturbation of a leg or joint position. In biological legged animals, such muscle length reflexes change the feedback sign from resistive during postural tasks (exploration behaviors mentioned by the authors) to assistive reflexes during locomotion (e.g. Edward, Prilutsky, In: Neurobiology of Motor Control: Fundamental Concepts and New Directions, New York: Wiley, 2017). The proposed network always inhibits the antagonist activity by mutual antagonistic inhibition to avoid the resistive reflexes. This seems to be an important issue for a discussion in the manuscript. By the way, a CPG appears to offer an elegant solution to the reflex sign change between resting and locomotor conditions – the same central rhythmic command excites synergists and inhibit antagonists (e.g., Geertsenet et al. J Physiol 589: 119-134, 2011).

3. The authors compare their computer simulations with recordings obtained in real hexapods. It is however not clear how differences in inertial leg properties between their simulated robot and real hexapods could affect the comparison. Inertia of real legs can be ignored (e.g. Hooper et al. J Neurosci 29: 4109-4119, 2009) but that of the robot, which is 20 times larger, probably not. Another difference on the design between the real hexapod and the robot is the presence of multi-joint muscles and passive tissues that couple leg joints; there are no similar structures in the robot.

4. The authors suggest that their distributed neural network model without CPG is novel. This claim may be inaccurate as somewhat similar models have been described in the literature and should be discussed in the manuscript (e.g. Geyer, Herr (2010) IEEE Trans.NeuralSyst.Rehabil.Eng. 18, 263–273; Dzeladini et al. Front Hum Neurosci 8: 371, 2014).

Specific comments

Line 22, “for starting or 22 interrupting a walk or changing velocity, all being difficult for CPG controlled solutions”: This statement does not seem accurate -- recent CPG models accurately predicts changing gaits with increasing locomotion speed (see Major comment 1 for references).

Lines 68-69, “But most of the cases studied concern swimming or flying or fast running”: Can the authors provide references for this statement, which does not seem accurate. For example, Yakovenko et al (J Neurophysiol 94: 1057-1065, 2005) have demonstrated a wide range of flexor and extensor burst activities in the cat fictive locomotion preparation corresponding to very slow walking to fast running in the cat.

Lines 139-141, “the fastest reflex response requires between 16 and 20 ms”: I understand that these numbers are taken from a published work, but the authors should keep in mind that this reflex response time is actually much longer than the fastest reflex in a much larger animal, the cat; the stretch reflex in cats has a delay of 8-10 ms.

Line 272, “velocity signals”: velocity-dependent signals?

Lines 278-279, “The joints of a leg are often not moved in phase, but, depending on the current environmental situation, may show varying phase values.”: Inter-joint coordination can be affected by multi-joint muscles and their passive properties and sensory feedback. This aspect of coordination is not discussed but appears to be important at least in mammalian locomotion (cheetahs, horses, humans).

Lines 336-338, “As an insect, the model has three pairs of legs, front legs, middle legs and hind legs…”: There is a critical difference in the inertial properties and thus dynamic behavior of robotic and insect legs -- the inertia of insect legs can be neglected (Hooper et al. 2009) but not in robot. It is important to discuss this difference and its effects on locomotor dynamics and control (see also Major comment 3).

Lines 384-390, “Using the traditional characterization…”: Which elements of the control structure prevents 2-, 1- and no limb support phases?

Line 613, “a decentralized controller”: This term is somewhat misleading as the network has "central rules" of control that are applied to all DOF.

Lines 615-616, “a detailed neuronal realization based on an antagonistic structure”: This organization appears to resemble the CPG organization based on the unit burst generators proposed by Grillner, 1981. It would be appropriate to reference this work and discuss the differences in organization between the neuroWalknet and unit burst generators.

Lines 758-760, “Furthermore, as no CPGs are applied on this level…”: Mutually inhibitory connections between the antagonist neurons may be considered CPG components as in the unit burst generators (Grillner, 1981). So the statement that 'no CPGs are applied to this level' may be somewhat misleading.

Line 771, “Levator Reflex”: It is not clear from the model description and Discussion how this and other motor reflexes are regulated by the network to ensure that postural reflexes do not interfere with the ongoing locomotion (e.g. Edwards, Prilutsky In Neurobiology of Motor Control (eds. Hooper, Buschges), 2017.

Lines 795-797, “These results provide an alternative explanation…”: As mentioned in the previous comments, a CPG does not have to consist of a central network setting a single rhythm and pattern (e.g., Grillner, 1981). On the other hand, the idea of locomotor control without CPG is not completely novel either -- see for example Geyer, Herr (2010) IEEE Trans.NeuralSyst.Rehabil.Eng. 18, 263–273; Dzeladini et al. Front Hum Neurosci 8: 371, 2014). That should be acknowledged in Discussion.

Line 808, “velocities of 10mV or less”: Sounds awkward’ consider changing e.g. to “Velocities corresponding to 10mV or less”.

Lines 865-868, “This suggests…”: In these situations the system also relies on the intrinsic stiffness properties of the legs to resist perturbations (e.g. Spagna et al. Bioinspir Biomim 2: 9-18, 2007).

Lines 881-884, “For more natural behaviors…”: This characterization of the role of sensory feedback in CPG-based models does not seem accurate (Major comment 1).

**Have all data underlying the figures and results presented in the manuscript been provided?**

Reviewer #1: Yes

Reviewer #2: Yes

Reviewer #3: Yes

Reviewer #4: Yes

Reviewer #5: Yes

PLOS authors have the option to publish the peer review history of their article (what does this mean?). If published, this will include your full peer review and any attached files.

Reviewer #1: No

Reviewer #2: Yes: Sasha N Zill

Reviewer #3: Yes: Wolfgang Stein

Reviewer #4: No

Reviewer #5: No

---

## [Decision Letter · Decision Letter 1]

3 Jan 2020

Dear Dr. Schilling,

Thank you very much for submitting your manuscript "Decentralized control of insect walking - a simple neural network explains a wide range of behavioral and neurophysiological results" (PCOMPBIOL-D-19-01265R1) for consideration at PLOS Computational Biology. As with all papers peer reviewed by the journal, your manuscript was reviewed by members of the editorial board and by several independent peer reviewers. Based on the reports, I regret to inform you that we will not be pursuing this manuscript for publication at PLOS Computational Biology.

 This editor and reviewer 1 have identified serious issues with the manuscript that were not adequately addressed in the revised manuscript. The review of the revision by reviewers 2-5 was, at best, perfunctory. The primary issue has to do with omitting the dynamics of the muscles, the body, gravity and other environmental perturbations. As the sensory feedback must result from muscle action this is an overriding issue. Moreover, this editor is especially concerned with the network presented in Fig. 2. As most of the elements are not even labeled it is not clear how their existence was established. Where is their occurrence and properties documented in the literature? 

Moreover, how were the detailed patterns of synaptic connectivity established experimentally and especially their relative strengths and reversal potentials. The lack of description as promised in the text and supplement (Line 332) leads to the conclusion that they represent conjecture. The ring network in Fig. 11 is at best even more fanciful conjecture.

Furthermore, the nature of the integrative neuronal modeling (lines 1043-1063) is highly questionable. The equation for Vupdate (line 1050) is a fanciful simplification. Where are the Ca++, NaP, K+, TRP currents and other forms of neuromodulation, etc. that are known to be inherent in rhythmic neurons? Instead of neuromodulation the authors provide the presence and absence of pilocarpine (a non-specific cholinergic agonist) as the means of turning the network on and off and its locus of action in the network is not specified. The time-based waveform for Vupdate and its transformation to the bar diagrams in the figures is not provided. The legs also participate in non-locomotory behavioral acts. How are functions like grooming and righting mediated? The bar diagrams are adequate to assess inter-appendage coordination but not individual limb kinematics. 

The arguments provided by the authors are simply inadequate to dismiss the central pattern generator as an underlying mechanism for the control of rhythmic locomotory behavior.

The reviews are attached below this email, and we hope you will find them helpful if you decide to revise the manuscript for submission elsewhere. I am sorry that we cannot be more positive on this occasion. We very much appreciate your wish to present your work in one of PLOS's Open Access publications. 

Thank you for your support, and I hope that you will consider PLOS Computational Biology for other submissions in the future.

Sincerely,

Joseph Ayers, PhD

Associate Editor

PLOS Computational Biology

Samuel Gershman

Deputy Editor

PLOS Computational Biology

Reviewer's Responses to Questions

**Comments to the Authors: **

Reviewer #1: In revising the manuscript the authors have altered their text thoroughly according to specific suggestions of the other four reviewers. In responding to the five major criticisms formulated in my initial review, they invested a lot of text to justify their approach, unfortunately their action not leading to any changes related to my feedback to the revised text.

Specifically, the authors formulate the challenge to introduce any changes related to my feedback, because the other reviewers did not state similar criticisms. I share this feeling to some basic extent as an author of scientific publications myself.

However, given the way, how I formulated my initial criticisms, i.e. scientifically based on the state of current knowledge in the field of motor control, I have some trouble to follow the arguments of the authors towards my initial concerns in the light of serving some majority vote. I have severe trouble to capture this notion in light of the task of scientists to move a specific field forward in understanding by means of scientific sound publications.

Given this situation I see it as task of the receiving editor to judge on the relevance of the reviewers comments to be relevant for a decision.

In the following, the review will give the relevant parts of the authors response marked by “>” with my comments stated below with “Rev:” preceding.

Response to my initial point 1:

Point 1: “We are aware that the manuscript is quite long in its current form. We tried to shorten (following explicit suggestions from reviewer 3) where possible. But, from our point of ………”

Rev: Still I am not comfortable with the concept of the authors in presenting their study, but in case the journal format of PLOS Computational Biology includes studies of that kind, I agree.

Response to my initial point 2:

Point 2: “Reviewer 1 criticizes that our dynamic simulation leaves the “most important

property unrepresented”, namely the relevance of muscle properties. Reviewer 1 is pointing out that there is a large number of nice simulation studies that focus on this aspect and can account well for these properties. We didn’t intend to downplay these important contributions and are fully aware of the relevance of muscles and their properties (we have therefore quoted the modeling approaches relevant for us in our Methods section, now lines 1477 – 1504). But we are simply focusing on a different level of description in our approach……….”

Rev: I agree fully with the notion of the authors to restrict their modelling work to a specific level of processing in motor control for locomotion, here the presumed neural level. However, given the fact that the authors aim to place their study within the state-of-the-art knowledge on one thoroughly studied model system, i.e. the stick insect, serving for most of the used results for modelling, it is of upmost scientific importance that they place their study in the hallmark of the currently available knowledge available on this animal. In doing so, they need to include reference to knowledge that was not included. Reference has therefore to be given to relevant publications on recent neuro-mechanical properties of the stick insect motor system, e.g. published by Blümel et al. 2012 (three publications), Hooper et al. 2006, 2007, 2009, and von Twickel et al. 2019.

Response to my initial point 3:

Point 3:” Another concern addresses the role of network components of the simulation (Introduction Responses to Reviewers – Decentralized control of insect walking 5 and Figure2): In the introduction the authors try hard to relativize the functional significance of the ubiquitous finding that central neural circuits generating locomotor activity in many (if not all) studied rhythmic locomotor forms, i.e. swimming, flying, walking, and crawling, contain elements or networks that can generate rhythmic activity without any phasic input, socalled CPGs.

In short, Reviewer 1 states that CPGs necessarily contribute to walking. This is exactly the statement that we are challenging (see below). We tried to clarify this in the introduction………………”

Rev: This statement is wrong. My initial review did not state, what the authors are quoting. CPGs are neural networks in the central nervous system, which can generate a rhythmic activity without any phasic external input, shown in some systems by pharmacological activation.

The authors incorrectly cite the locust flight system as an example for a motor system driven by CPG activity: in contrast to the authors’ statement, the locust flight system was the first, for which Pearson and colleagues have shown that it is sensory feedback (e.g. Pearson & Wolf, 1988; Pearson & Ramirez, 1990 and further articles), which affects and sculpts the activity of a centrally generated motor activity by impinging on the network in a way that the CPG only serves in providing the network topology for the externally induced information to act on.

The same holds for results from previous studies on walking in insects, in which sensory feedback from the legs was shown to use central neural network topology in contributing to the stepping motor output (e.g Akay et al. 2007; Bucher et al. 2003; Hess & Büschges, 1999).

Response to my initial point 4:

Point 4: “Reviewer 1 states that we do not present quantitative data except for Fig 5. We disagree with this view. From our point of view, the figures show quantitative data, as does figure 5 (in particular, figure 5 represents an abstracted version of figure 3 and a summary of the results of that figure)………….”

Rev: The reviewer still emphasizes that figures 3, 4, 6, 7, 8 and 9 in the revised version of the manuscript give examples of th

Reviewer #2: The authors have addressed all my comments and suggestions. I strongly recommend that this important paper be published.

Reviewer #3: The authors have responded to all my comments and edited the manuscript appropriately. I have only a few minor suggestions left, none of which should delay the publication of the paper. I use the line numbers from the annotated pdf. 

490 – 530: Maybe a bit too wordy. The sentences could be more concise. 

529 "Tests with varying parameters showed that application of different local velocities was indeed necessary to elicit this kind of curve walking. Changing only the AEP of front legs was not sufficient". I am unclear about how necessity and sufficiency were tested here. 

822. What is meant by 'there are no experimental results concerning I1'? This statement is either too broad or the context is not clear. 

890 Change to "does not oscillate EITHER.'

962 and other places. Like one of the other reviewers mentioned, "mv" for velocity appears unusual. 

1001: 'as addressed in Results' makes no sense since this is now in the results section

1371: The writing of the new section starting in line 1368 seems a bit clunky and could be shortened. For example "As detailed data are available for one specific case only, we could not aim for a general solution, but, in order to approximate data of Dürr and Ebeling (2005), chose plausible set point values for". Did you mean "As detailed experimental data are only available from Dürr and Ebeling (2005), we chose plausible set point values that approximated..... ?

Similarly, in line 1380 "As there is no detailed information given concerning the outer leg" It would be better to make clear that Dürr and Ebeling do not give this information (vs you not having that information) and reformulate the sentence to make it simpler. 

1385 "Again, as only this data set is available, we did not attempt to find a general solution for all possible turning radii, but focused on this special case. " Sounds very repetitive (maybe because you were trying to satisfy another reviewer).

Reviewer #4: I am satisfied with the changes the authors made to the manuscript and recommend acceptance.

Reviewer #5: The authors addressed all my comments satisfactorily and revised the manuscript accordingly. I have no further comments.

**Have all data underlying the figures and results presented in the manuscript been provided?**

Reviewer #1: None

Reviewer #2: Yes

Reviewer #3: Yes

Reviewer #4: Yes

Reviewer #5: Yes

PLOS authors have the option to publish the peer review history of their article (what does this mean?). If published, this will include your full peer review and any attached files.

Reviewer #1: No

Reviewer #2: Yes: Sasha N Zill

Reviewer #3: Yes: Wolfgang Stein

Reviewer #4: No

Reviewer #5: No

---

## [Decision Letter · Decision Letter 2]

19 Mar 2020

Dear Dr. Schilling,

We are pleased to inform you that your manuscript 'Decentralized control of insect walking - a simple neural network explains a wide range of behavioral and neurophysiological results' has been provisionally accepted for publication in PLOS Computational Biology.

Best regards,

Samuel J. Gershman

Deputy Editor

PLOS Computational Biology

Reviewer's Responses to Questions

**Comments to the Authors:**

Reviewer #1: The authors have submitted a revised version of their manuscript. Differing from previous revision the authors have added and/or changed specific parts of their text to address the aspects of my criticisms. Unfortunately, the line numbers quoted in the rebuttle letter do not match the line numbers of the uploaded manuscript. Even though, I have generated this review as thoroughly as possible asking for long lasting searches for the alterations referred to in the authors’ rebuttle letter.

The revision now satisfactorily includes modifications that I stimulated with my previous comments, e.g. additional data and description on forward walking, backward walking and the specifics of activation of load sensors. As well and at least the authors included their view on the role of muscular and biomechanical properties for simulations from their perspective to the text.

**Have all data underlying the figures and results presented in the manuscript been provided?**

Reviewer #1: Yes

PLOS authors have the option to publish the peer review history of their article (what does this mean?). If published, this will include your full peer review and any attached files.

Reviewer #1: No

---

## [Editor Report · Acceptance letter]

13 Apr 2020

PCOMPBIOL-D-19-01265R2 

Decentralized control of insect walking - a simple neural network explains a wide range of behavioral and neurophysiological results

Dear Dr Schilling,

I am pleased to inform you that your manuscript has been formally accepted for publication in PLOS Computational Biology. Your manuscript is now with our production department and you will be notified of the publication date in due course.

With kind regards,

Laura Mallard
